# Koopa: Learning Non-stationary Time Series Dynamics with Koopman Predictors

**Yong Liu**,* **Chenyu Li**,* **Jianmin Wang, Mingsheng Long**$^{\boxtimes}$
School of Software, BNRist, Tsinghua University, China
{liuyong21,lichenyu20}@mails.tsinghua.edu.cn, {jimwang,mingsheng}@tsinghua.edu.cn

## Abstract

Real-world time series are characterized by intrinsic non-stationarity that poses a principal challenge for deep forecasting models. While previous models suffer from complicated series variations induced by changing temporal distribution, we tackle non-stationary time series with modern Koopman theory that fundamentally considers the underlying time-variant dynamics. Inspired by Koopman theory that portrays complex dynamical systems, we disentangle time-variant and time-invariant components from intricate non-stationary series by *Fourier Filter* and design *Koopman Predictor* to advance respective dynamics forward. Technically, we propose **Koopa** as a novel **Koop**man forec**a**ster composed of stackable blocks that learn hierarchical dynamics. Koopa seeks measurement functions for Koopman embedding and utilizes Koopman operators as linear portraits of implicit transition. To cope with time-variant dynamics that exhibits strong locality, Koopa calculates context-aware operators in the temporal neighborhood and is able to utilize incoming ground truth to scale up forecast horizon. Besides, by integrating Koopman Predictors into deep residual structure, we ravel out the binding reconstruction loss in previous Koopman forecasters and achieve end-to-end forecasting objective optimization. Compared with the state-of-the-art model, Koopa achieves competitive performance while saving 77.3% training time and 76.0% memory. Code is available at this repository: https://github.com/thuml/Koopa.

## 1 Introduction

Time series forecasting has become an essential part of real-world applications, such as weather forecasting, energy consumption, and financial assessment. With numerous available observations, deep learning approaches exhibit superior performance and bring the boom of deep forecasting models. TCNs [4, 43, 47] utilize convolutional kernels and RNNs [12, 22, 37] leverage the recurrent structure to capture underlying temporal patterns. Afterward, attention mechanism [42] becomes the mainstream of sequence modeling and Transformers [31, 48, 53] show great predictive power with the capability of learning point-wise temporal dependencies. And the recent revival of MLPs [32, 51, 52] presents a simple but effective approach to exhibit temporal dependencies by dense weighting.

In spite of elaboratively designed models, it is a fundamental problem for deep models to generalize on varying distribution [1, 25, 33], which is widely reflected in real-world time series because of inherent non-stationarity. Non-stationary time series is characterized by time-variant statistics and temporal dependencies in different periods [2, 14], inducing a huge distribution gap between training and inference and even among each lookback window. While previous methods [16, 28] tailor existing architectural design to attenuate the adverse effect of non-stationarity, few works research on the theoretical basis that can be applied to deal with time-variant temporal patterns naturally.

---

*Equal Contribution

37th Conference on Neural Information Processing Systems (NeurIPS 2023).

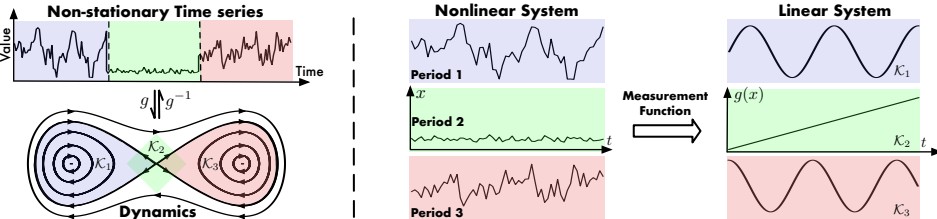

Figure 1: The measurement function $g$ maps between non-stationary time series and the nonlinear dynamical system so that the timeline will correspond to a system trajectory. Therefore, time series variations in different periods are reflected as sub-regions of nonlinear dynamics, which can be portrayed and advanced forward in time by linear Koopman operators $\{\mathcal{K}_1, \mathcal{K}_2, \mathcal{K}_3\}$ respectively.

From another perspective, real-world time series acts like time-variant dynamics [6]. As one of the principal approaches to analyze complex dynamics, Koopman theory [20] provides an enlightenment to transform nonlinear system into measurement function space, which can be described by a linear Koopman operator. Several pilot works accomplish the integration with deep learning approaches by employing autoencoder networks [40] and operator-learning [26, 49]. More importantly, it is supported by Koopman theory that for time-variant dynamics, there exists a coordinate transformation of the system, where localized Koopman operators are valid to describe the whole measurement function space into several subspaces with linearization [23, 36]. Therefore, Koopman-based methods are appropriate to learn non-stationary time series dynamics (Figure 1). Besides, the linearity of measurement function space enables us to utilize spectral analysis to interpret nonlinear systems.

In this paper, we disentangle non-stationary series into time-invariant and time-variant dynamics and propose **Koopa** as a novel **Koop**man fore**ca**ster, which is composed of modular **K**oopman **P**redictors (**KP**) to hierarchically describe and advance forward series dynamics. Concretely, we utilize Fourier analysis for dynamics disentangling. And for time-invariant dynamics, the model learns Koopman embedding and linear operators to reveal the implicit transition underlying long-term series. As for the remaining time-variant components that exhibit strong locality, Koopa performs context-aware operator calculation and adaptation within different lookback windows. Besides, Koopman Predictor goes beyond the canonical design of Koopman Autoencoder without the binding reconstruction loss, and we incorporate modular blocks into deep residual architecture [32] to realize end-to-end time series forecasting. Our contributions are summarized as follows:

- From the perspective of modern dynamics Koopman theory, we propose *Koopa* composed of modular *Fourier Filter* and *Koopman Predictor*, which can hierarchically disentangle and exploit time-invariant and time-variant dynamics for time series forecasting.
- Based on the linearity of Koopman operators, the proposed model is able to utilize incoming series and adapt to varying dynamics for scaling up forecast horizon.
- Compared with state-of-the-art methods, our model achieves competitive performance while saving 77.3% training time and 76.0% memory averaged from six real-world benchmarks.

## 2 Related Work

### 2.1 Time Series Forecasting with DNNs

Deep neural networks (DNNs) have made great breakthroughs in time series forecasting. TCN-based models [4, 43, 47] explore hierarchical temporal patterns and adopt shared convolutional kernels with diverse receptive fields. RNN-based models [12, 22, 37] utilize the recurrent structure with memory to reveal the implicit transition over time points. MLP-based models [32, 51, 52] learn point-wise weighting and the impressive performance and efficiency highlight that MLP performs well for modeling simple temporal dependencies. However, their practical applicability may still be constrained on non-stationary time series, which is endowed with time-variant properties and poses challenges for model capacity and efficiency. Unlike previous methods, Koopa fundamentally considers the complicated dynamics underlying time series and implements efficient and interpretable transition learners in both time-variant and time-invariant manners inspired by Koopman theory.

Recently, Transformer-based models have also achieved great success in time series forecasting. Initial attempts [19, 27, 48, 53] renovate the canonical structure and reduce the quadratic complexity

for long-term forecasting. However, recent studies [16, 21] find it a central problem for Transformer and other DNNs to generalize on varying temporal distribution and several works [15, 16, 21, 28] tailor to empower the robustness against shifted distribution. Especially, PatchTST [31] boosts Transformer to the state-of-the-art performance by channel-independence and instance normalization [41] but may lead to unaffordable computational cost when the number of series variate is large. In this paper, our proposed model supported by Koopman theory works naturally for non-stationary time series and achieves the state-of-the-art forecasting performance with remarkable model efficiency.

### 2.2 Learning Dynamics with Koopman Operator

Koopman theory [20] has emerged over decades as the dominant perspective to analyze modern dynamical systems [6]. Together with Dynamic Mode Decomposition (DMD) [38] as the leading numerical method to approximate the Koopman operator, significant advances have been accomplished in aerodynamics and fluid physics [3, 9, 30]. Recent progress made in Koopman theory is inherently incorporated with deep learning approaches in the data science era. Pilot works [29, 40, 50] leverage data-driven approaches such as Koopman Autoencoder to learn the measurement function and operator simultaneously. PCL [3] further introduces a backward procedure to improve the consistency and stability of the operator. Based on the capability of Koopman operator to advance nonlinear dynamics forward, it is also widely applied to sequence prediction. By means of Koopman spectral analysis, MDKAE [5] disentangles dominant factors underlying sequential data and is competent to forecast with specific factors. K-Forecast [24] utilizes Koopman theory to handle nonlinearity in temporal signals and propose to optimize data-dependent basis for long-term time series forecasting. By leveraging predefined measurement functions, KNF [44] learns Koopman operator and attention map to cope with time series forecasting with changing temporal distribution.

Different from previous Koopman forecasters, we design modular Koopman Predictors to tackle time-variant and time-invariant components with hierarchically learned operators, and renovate Koopman Autoencoder by removing the reconstruction loss to achieve fully predictive training.

## 3 Background

### 3.1 Koopman Theory

A discrete-time dynamical system can be formulated as $x_{t+1} = \mathbf{F}(x_t)$, where $x_t$ denotes the system state and $\mathbf{F}$ is a vector field describing the dynamics. However, it is challenging to identify the system transition directly on the state because of nonlinearity or noisy data. Instead, Koopman theory [20] hypothesizes the state can be projected into the space of measurement function $g$, which can be governed and advanced forward in time by an infinite-dimensional linear operator $\mathcal{K}$, such that:

$$\mathcal{K} \circ g(x_t) = g\big(\mathbf{F}(x_t)\big) = g(x_{t+1}). \tag{1}$$

Koopman theory provides a bridge between finite-dimensional nonlinear dynamics and infinite-dimensional linear dynamics, where spectral analysis tools can be applied to obtain in-depth analysis.

### 3.2 Dynamic Mode Decomposition

Dynamic Mode Decomposition (DMD) [38] seeks the best fitted finite-dimensional matrix $K$ to approximate infinite-dimensional operator $\mathcal{K}$ by collecting the observed system state (a.k.a. *snapshot*). Although DMD is the standard numerical method to analyze dynamics, it only works on linear space assumptions, which can be hardly identified without prior knowledge. Therefore, eDMD [45] is proposed to avoid handcrafting measurement functions and harmonious incorporations are made with the learning approach by employing autoencoders, which yields Koopman Autoencoder (KAE). By the universal approximation theorem [13] of deep networks, KAE finds desired *Koopman embedding* $g(x_t)$ with learned measurement function in a data-driven approach.

### 3.3 Time Series as Dynamics

It is challenging to predict real-world time series because of inherent non-stationarity. But if we zoom in the timeline, we will find the localized time series exhibited weak stationarity. It coincides with Koopman theory to analyze large nonlinear dynamics. That is, the measurement

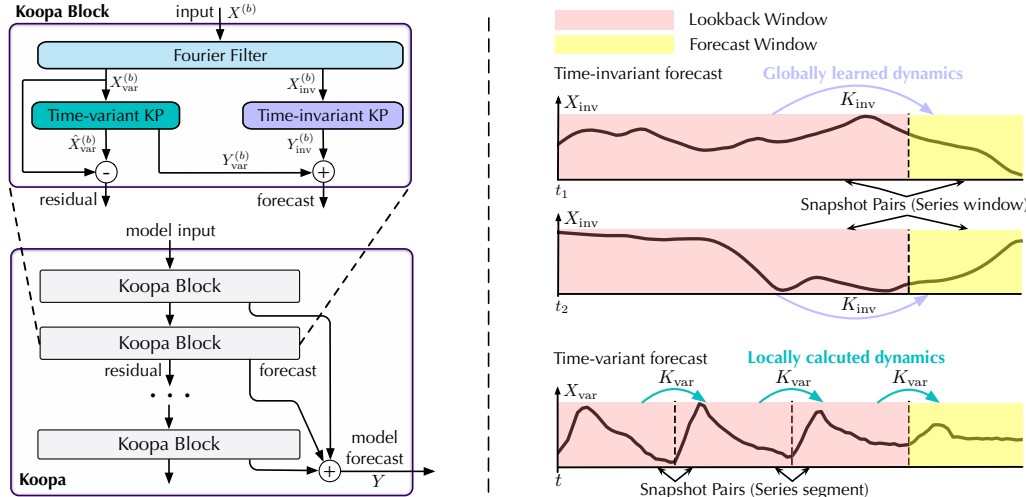

Figure 2: Left: Koopa structure. By taking the residual of previous block fitted dynamical system, each block learns hierarchical dynamics and Koopa aggregates the forecast of all blocks. Right: For time-invariant forecast, Koopa learns globally shared dynamics from each lookback-forecast window pair. For time-variant forecast, the model calculates localized and segment-wise dynamics.

function space can be divided into several neighborhoods, which are discriminately portrayed by localized linear operators [23]. Therefore, we leverage Koopman-based approaches that tackle large nonlinear dynamics by disentangling time-variant and time-invariant dynamics. Inspired by Wold's Theorem [46] that every covariance-stationary time series $X_t$ can be formally decomposed as:

$$X_t = \eta_t + \sum_{j=0}^{\infty} b_j \varepsilon_{t-j}, \tag{2}$$

where $\eta_t$ denotes the deterministic component and $\varepsilon_t$ is the stochastic component as the stationary process input of linear filter $\{b_j\}$, we introduce globally learned and localized linear Koopman operators to exploit respective dynamics underlying different components.

## 4 Koopa

We propose *Koopa* composed of stackable *Koopa Blocks* (Figure 2). Each block is obliged to learn the input dynamics and advance it forward for prediction. Instead of struggling to seek one unified operator that governs the whole measurement function space, each Koopa Block is encouraged to learn operators hierarchically by taking the residual of previous block fitted dynamics as its input.

**Koopa Block** As aforementioned, it is essential to disentangle different dynamics and adopt proper operators for non-stationary series forecasting. The proposed block shown in Figure 2 contains *Fourier Filter* that utilizes frequency domain statistics to disentangle time-variant and time-invariant components and implements two types of *Koopman Predictor (KP)* to obtain Koopman embedding respectively. In Time-invariant KP, we set the operator as a model parameter to be globally learned from lookback-forecast windows. In Time-variant KP, analytical operator solutions are calculated locally within the lookback window, with series segments arranged as snapshots. In detail, we formulate the $b$-th block input $X^{(b)}$ as $[x_1, x_2, \dots, x_T]^\top \in \mathbb{R}^{T \times C}$, where $T$ and $C$ denote the lookback window length and the variate number. The target is to output a forecast window of length $H$. Our proposed Fourier Filter conducts disentanglement at the beginning of each block:

$$X_{\text{var}}^{(b)}, \; X_{\text{inv}}^{(b)} = \text{FourierFilter}(X^{(b)}). \tag{3}$$

Respective KPs will predict with time-invariant input $X_{\text{inv}}^{(b)}$ and time-variant input $X_{\text{var}}^{(b)}$, and Time-variant KP simultaneously outputs the fitted input $\hat{X}_{\text{var}}^{(b)}$:

$$\begin{aligned} Y_{\text{inv}}^{(b)} &= \text{TimeInvKP}(X_{\text{inv}}^{(b)}), \\ \hat{X}_{\text{var}}^{(b)}, Y_{\text{var}}^{(b)} &= \text{TimeVarKP}(X_{\text{var}}^{(b)}). \end{aligned} \tag{4}$$

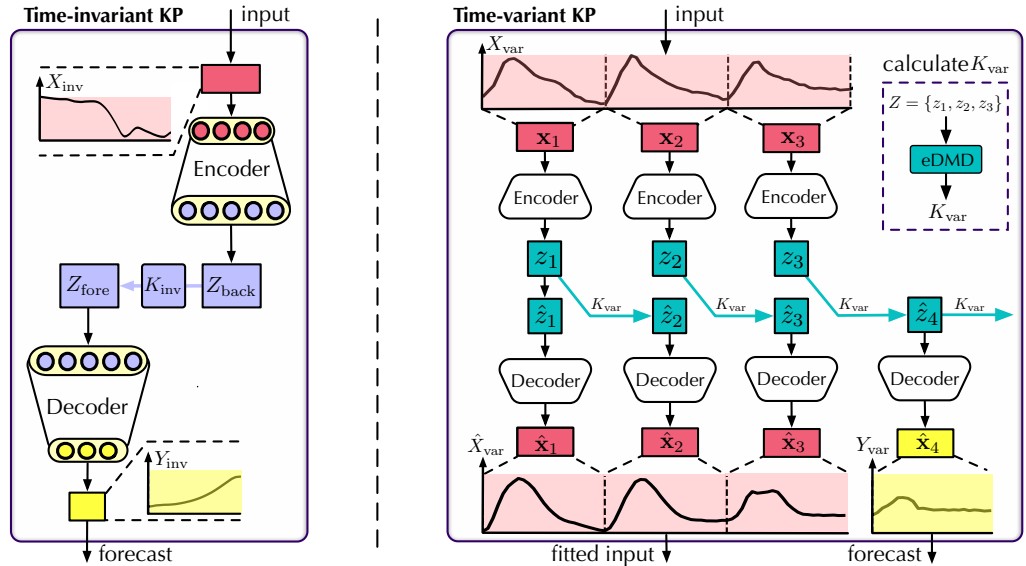

Figure 3: Left: Time-invariant KP learns Koopman embedding and operator with time-invariant components globally from all windows. Right: Time-variant KP conducts localized operator calculation within lookback window and advances dynamics forward with the obtained operator for predictions.

Unlike KAEs [45, 44] that introduce a loss term for rigorous reconstruction of the lookback-window series, we feed the residual $X^{(b+1)}$ as the input of next block for learning a corrective operator. And the model forecast $Y$ is the sum of predicted components $Y_{\text{var}}^{(b)}, Y_{\text{inv}}^{(b)}$ gathered from all Koopa Blocks:

$$X^{(b+1)} = X_{\text{var}}^{(b)} - \hat{X}_{\text{var}}^{(b)}, \quad Y = \sum \left( Y_{\text{var}}^{(b)} + Y_{\text{inv}}^{(b)} \right). \tag{5}$$

**Fourier Filter** To disentangle the series components, we leverage Fourier analysis to find the globally shared and localized frequency spectrums reflected on different periods. Concretely, we precompute the Fast Fourier Transform (FFT) of each lookback window of the training set, calculate the averaged amplitude of each spectrum $\mathcal{S} = \{0, 1, \dots, [T/2]\}$, and sort them by corresponding amplitude. We take the top percent of $\alpha$ as the subset $\mathcal{G}_\alpha \subset \mathcal{S}$, which contains dominant spectrums shared among all lookback windows and exhibits time-invariant dynamics underlying the dataset. And the remaining spectrums are the specific ingredient for varying windows in different periods. Therefore, we divide the spectrums $\mathcal{S}$ into $\mathcal{G}_\alpha$ and its complementary set $\bar{\mathcal{G}}_\alpha$. During training and inference, FourierFilter($\cdot$) conducts the disentanglement of input $X$ (block superscript omitted) as

$$\begin{aligned} X_{\text{inv}} &= \mathcal{F}^{-1}\big( \text{Filter}\left(\mathcal{G}_\alpha, \ \mathcal{F}(X)\right)\big), \\ X_{\text{var}} &= \mathcal{F}^{-1}\big( \text{Filter}\left(\bar{\mathcal{G}}_\alpha, \ \mathcal{F}(X)\right)\big) = X - X_{\text{inv}}, \end{aligned} \tag{6}$$

where $\mathcal{F}$ means FFT, $\mathcal{F}^{-1}$ is its inverse and Filter($\cdot$) only passes corresponding frequency spectrums with the given set. We validate the disentangling effect of our proposed Fourier Filter in Section 5.2 by calculating the variation degree of temporal dependencies in the disentangled series.

**Time-invariant KP** Time-invariant KP is designed to portray the globally shared dynamics, which discovers the direct transition from lookback window to forecast window as $\mathbf{F} : X_{\text{inv}} \mapsto Y_{\text{inv}}$. Concretely, we introduce a pair of Encoder : $\mathbb{R}^{T \times C} \mapsto \mathbb{R}^D$ and Decoder : $\mathbb{R}^D \mapsto \mathbb{R}^{H \times C}$ to learn the common Koopman embedding for the time-invariant components of running window pairs, where $D$ denotes the embedding dimension. Working on the data-driven measurement function, we introduce the operator $K_{\text{inv}} \in \mathbb{R}^{D \times D}$ as a learnable parameter in each Time-invariant KP, which regards the embedding of lookback and forecast window $Z_{\text{back}}, Z_{\text{fore}} \in \mathbb{R}^D$ as running snapshot pairs. The procedure is shown in Figure 3 and TimeInvKP($\cdot$) is formulated as follows:

$$Z_{\text{back}} = \text{Encoder}(X_{\text{inv}}), \ Z_{\text{fore}} = K_{\text{inv}} Z_{\text{back}}, \ Y_{\text{inv}} = \text{Decoder}(Z_{\text{fore}}). \tag{7}$$

**Time-variant KP**  As time-variant dynamics changes continuously, we utilize localized snapshots in a window, which constitute a temporal neighborhood more likely to be linearized. To obtain semantic snapshots and reduce iterations, the input $X_{\text{var}}$ is divided into $\frac{T}{S}$ segments $\mathbf{x}_j$ of length $S$:

$$\mathbf{x}_j = [x_{(j-1)S+1}, \ldots, x_{jS}]^\top \in \mathbb{R}^{S \times C}, \; j = 1, 2, \ldots, T/S. \tag{8}$$

We assume $S$ is divisible by $T$ and $H$; otherwise, we pad the input or truncate the output to make it compatible. Time-variant KP aims to portray localized dynamics, which is manifested analytically as the segment-wise transition $\mathbf{F} : \mathbf{x}_t \mapsto \mathbf{x}_{t+1}$ with observed snapshots. We utilize another pair of Encoder : $\mathbb{R}^{S \times C} \mapsto \mathbb{R}^D$ to transform each segment into Koopman embedding $z_j$ and Decoder : $\mathbb{R}^D \mapsto \mathbb{R}^{S \times C}$ to transform the fitted or predicted embedding $\hat{z}_j$ back to time segments $\hat{\mathbf{x}}_j$:

$$z_j = \text{Encoder}(\mathbf{x}_j), \; \hat{\mathbf{x}}_j = \text{Decoder}(\hat{z}_j). \tag{9}$$

Given snapshots collection $Z = [z_1, \ldots, z_{\frac{T}{S}}] \in \mathbb{R}^{D \times \frac{T}{S}}$, we leverage eDMD [45] to find the best fitted matrix that advances forward the system. We apply one-step operator approximation as follows:

$$Z_{\text{back}} = [z_1, z_2, \ldots, z_{\frac{T}{S}-1}], \; Z_{\text{fore}} = [z_2, z_3, \ldots, z_{\frac{T}{S}}], \; K_{\text{var}} = Z_{\text{fore}} Z_{\text{back}}^\dagger, \tag{10}$$

where $Z_{\text{back}}^\dagger \in \mathbb{R}^{(\frac{T}{S}-1) \times D}$ is the Moore–Penrose inverse of lookback window embedding collection. The calculated $K_{\text{var}} \in \mathbb{R}^{D \times D}$ varies with windows and helps to analyze local temporal variations as a linear system. With the calculated operator, the fitted embedding is formulated as follows:

$$[\hat{z}_1, \hat{z}_2, \ldots, \hat{z}_{\frac{T}{S}}] = [z_1, K_{\text{var}} z_1, \ldots, K_{\text{var}} z_{\frac{T}{S}-1}] = [z_1, K_{\text{var}} Z_{\text{back}}]. \tag{11}$$

To obtain a prediction of length $H$, we iterate operator forwarding to get $\frac{H}{S}$ predicted embedding:

$$\hat{z}_{\frac{T}{S}+t} = (K_{\text{var}})^t z_{\frac{T}{S}}, \; t = 1, 2, \ldots, H/S. \tag{12}$$

Finally, we arrange the segments transformed by $\text{Decoder}(\cdot)$ as the module outputs $\hat{X}_{\text{var}}, Y_{\text{var}}$. The whole procedure is shown in Figure 3 and $\text{TimeVarKP}(\cdot)$ can be formulated as Equation 8–13.

$$\hat{X}_{\text{var}} = [\hat{\mathbf{x}}_1, \ldots, \hat{\mathbf{x}}_{\frac{T}{S}}]^\top, \; Y_{\text{var}} = [\hat{\mathbf{x}}_{\frac{T}{S}+1}, \ldots, \hat{\mathbf{x}}_{\frac{T}{S}+\frac{H}{S}}]^\top. \tag{13}$$

**Forecasting Objective**  In Koopa, Encoder, Decoder and $K_{\text{inv}}$ are learnable parameters, while $K_{\text{var}}$ is calculated on-the-fly. To maintain the Koopman embedding consistency in different blocks, we share Encoder, Decoder in Time-variant and Time-invariant KPs, which are formulated as $\phi_{\text{var}}$ and $\phi_{\text{inv}}$ respectively, and use the MSE loss with the ground truth $Y_{\text{gt}}$ for parameter optimization:

$$\text{argmin}_{K_{\text{inv}}, \phi_{\text{var}}, \phi_{\text{inv}}} \mathcal{L}_{\text{MSE}}(Y, Y_{\text{gt}}). \tag{14}$$

Optimizing by a single forecasting objective based on the assumption that if reconstruction failed, the prediction must also fail. Thus eliminating forecast discrepancy helps for fitting observed dynamics.

## 5   Experiments

**Datasets**  We conduct extensive experiments to evaluate the performance and efficiency of Koopa. For multivariate forecasting, we include six real-world benchmarks used in Autoformer [48]: ECL (UCI), ETT [53], Exchange [22], ILI (CDC), Traffic (PeMS), and Weather (Wetterstation). For univariate forecasting, we evaluate the performance on the well-acknowledged M4 dataset [39], which contains four subsets of periodically collected univariate marketing data. And we follow the data processing and split ratio used in TimesNet [47].

Notably, instead of setting a fixed lookback window length, for every forecast window length $H$, we set the length of lookback window $T = 2H$ as the same with N-BEATS [32], because historical observations are always available in real-world scenarios and it can be beneficial for deep models to leverage more observed data with the increasing forecast horizon.

**Baselines**  We extensively compare Koopa with the state-of-the-art deep forecasting models, including Transformer-based model: Autoformer [48], PatchTST [31]; TCN-based model: TimesNet [47], MICN [43]; MLP-based model: DLinear [51]; Fourier forecaster: FiLM [54], and Koopman forecaster: KNF [44]. We also introduce additional specialized models N-HiTS [7] and N-BEATS [32] for univariate forecasting as competitive baselines. All the baselines we reproduced are implemented based on the original paper or official code. We repeat each experiment three times with different random seeds and report the test MSE/MAE. And we provide detailed code implementation and hyperparameters sensitivity in Appendix C.

## 5.1 Time Series Forecasting

**Forecasting results**   We list the results in Table 1–2 with the best in **bold** and the second underlined. Koopa shows competitive forecasting performance in both multivariate and univariate forecasting. Concretely, Koopa achieves state-of-the-art performance in more than **70%** multivariate settings and consistently outperforms other deep models in the univariate settings.

Notably, Koopa surpasses the state-of-the-art Koopman-based forecaster KNF by a large margin in real-world time series, which can be attributed to our hierarchical dynamics learning and disentangling mechanism. Also, as the representative of efficient linear models, the performance of DLinear is still subpar in ILI, Traffic and Weather, indicating that nonlinear dynamics underlying the time series poses challenges for model capacity and point-wise weighting may not be appropriate to portray time-variant dynamics. Besides, compared with painstakingly trained PatchTST with channel-independence mechanism, our model can achieve a close and even better performance with naturally addressed non-stationary properties of real-world time series.

Table 1: Multivariate forecasting results with different forecast lengths $H \in \{24, 36, 48, 60\}$ for ILI and $H \in \{48, 96, 144, 192\}$ for others. We set the lookback length $T = 2H$. Additional results (ETTm1, ETTm2, ETTh1) are provided in Appendix D.1.

| Models | | **Koopa** | | PatchTST | | TimesNet | | DLinear | | MICN | | KNF | | FiLM | | Autoformer | |
|---|---|---|---|---|---|---|---|---|---|---|---|---|---|---|---|---|---|
| Metric | | MSE | MAE | MSE | MAE | MSE | MAE | MSE | MAE | MSE | MAE | MSE | MAE | MSE | MAE | MSE | MAE |
| ECL | 48 | **0.130** | **0.234** | 0.147 | 0.246 | 0.149 | 0.254 | 0.158 | 0.241 | 0.156 | 0.271 | 0.175 | 0.265 | 0.197 | 0.270 | 0.164 | 0.272 |
| | 96 | **0.136** | **0.236** | 0.143 | 0.241 | 0.170 | 0.275 | 0.153 | 0.245 | 0.165 | 0.277 | 0.198 | 0.284 | 0.238 | 0.341 | 0.182 | 0.289 |
| | 144 | 0.149 | 0.247 | **0.145** | **0.241** | 0.183 | 0.287 | 0.152 | 0.245 | 0.163 | 0.274 | 0.204 | 0.297 | 0.234 | 0.338 | 0.210 | 0.315 |
| | 192 | 0.156 | 0.254 | **0.147** | **0.240** | 0.189 | 0.291 | 0.153 | 0.246 | 0.171 | 0.284 | 0.245 | 0.321 | 0.240 | 0.339 | 0.221 | 0.324 |
| ETTh2 | 48 | 0.226 | **0.300** | **0.223** | **0.297** | 0.241 | 0.319 | 0.226 | 0.305 | 0.260 | 0.336 | 0.385 | 0.376 | 0.261 | 0.324 | 0.355 | 0.380 |
| | 96 | 0.297 | **0.349** | 0.300 | 0.353 | 0.325 | 0.376 | **0.294** | 0.351 | 0.343 | 0.393 | 0.433 | 0.446 | 0.322 | 0.372 | 0.427 | 0.432 |
| | 144 | **0.333** | **0.381** | 0.346 | 0.390 | 0.374 | 0.408 | 0.354 | 0.397 | 0.374 | 0.411 | 0.441 | 0.456 | 0.352 | 0.397 | 0.457 | 0.461 |
| | 192 | **0.356** | **0.393** | 0.383 | 0.406 | 0.394 | 0.434 | 0.385 | 0.418 | 0.455 | 0.464 | 0.528 | 0.503 | 0.361 | 0.410 | 0.503 | 0.491 |
| Exchange | 48 | **0.042** | **0.143** | 0.044 | 0.144 | 0.059 | 0.172 | 0.043 | 0.145 | 0.117 | 0.248 | 0.128 | 0.271 | 0.071 | 0.192 | 0.125 | 0.252 |
| | 96 | **0.083** | 0.207 | 0.085 | **0.204** | 0.120 | 0.255 | 0.084 | 0.220 | 0.108 | 0.251 | 0.294 | 0.394 | 0.112 | 0.245 | 0.280 | 0.386 |
| | 144 | **0.130** | 0.261 | 0.132 | 0.260 | 0.206 | 0.334 | 0.132 | **0.253** | 0.152 | 0.301 | 0.597 | 0.578 | 0.174 | 0.306 | 0.520 | 0.523 |
| | 192 | 0.184 | 0.309 | **0.174** | 0.300 | 0.377 | 0.463 | 0.178 | **0.299** | 0.187 | 0.331 | 0.654 | 0.595 | 0.241 | 0.364 | 0.653 | 0.592 |
| ILI | 24 | **1.621** | **0.800** | 2.063 | 0.881 | 2.464 | 1.039 | 2.624 | 1.118 | 4.380 | 1.558 | 3.722 | 1.432 | 3.590 | 1.424 | 2.831 | 1.085 |
| | 36 | **1.803** | **0.855** | 2.178 | 0.943 | 2.388 | 1.007 | 2.693 | 1.156 | 3.314 | 1.313 | 3.941 | 1.448 | 4.200 | 1.383 | 2.801 | 1.088 |
| | 48 | **1.768** | 0.903 | 1.916 | **0.896** | 2.370 | 1.040 | 2.852 | 1.229 | 2.457 | 1.085 | 3.287 | 1.377 | 3.317 | 1.417 | 2.322 | 1.006 |
| | 60 | **1.743** | **0.891** | 1.981 | 0.917 | 2.193 | 1.003 | 2.554 | 1.144 | 2.379 | 1.040 | 2.974 | 1.301 | 4.077 | 1.444 | 2.470 | 1.061 |
| Traffic | 48 | **0.415** | **0.274** | 0.426 | 0.286 | 0.567 | 0.306 | 0.488 | 0.352 | 0.496 | 0.301 | 0.621 | 0.382 | 0.498 | 0.312 | 0.640 | 0.361 |
| | 96 | **0.401** | **0.275** | 0.413 | 0.283 | 0.611 | 0.337 | 0.485 | 0.336 | 0.511 | 0.312 | 0.645 | 0.376 | 0.451 | 0.297 | 0.668 | 0.367 |
| | 144 | **0.397** | **0.276** | 0.405 | 0.278 | 0.603 | 0.322 | 0.452 | 0.317 | 0.498 | 0.309 | 0.683 | 0.402 | 0.430 | 0.288 | 0.681 | 0.379 |
| | 192 | **0.403** | 0.284 | 0.404 | **0.277** | 0.604 | 0.321 | 0.438 | 0.309 | 0.494 | 0.312 | 0.699 | 0.405 | 0.425 | 0.288 | 0.692 | 0.385 |
| Weather | 48 | **0.126** | **0.168** | 0.140 | 0.179 | 0.138 | 0.191 | 0.156 | 0.198 | 0.157 | 0.217 | 0.201 | 0.288 | 0.160 | 0.206 | 0.185 | 0.240 |
| | 96 | **0.154** | **0.205** | 0.160 | 0.206 | 0.180 | 0.231 | 0.186 | 0.229 | 0.187 | 0.250 | 0.295 | 0.308 | 0.189 | 0.233 | 0.230 | 0.279 |
| | 144 | **0.172** | 0.225 | 0.174 | **0.221** | 0.190 | 0.244 | 0.199 | 0.244 | 0.197 | 0.257 | 0.394 | 0.401 | 0.200 | 0.245 | 0.268 | 0.308 |
| | 192 | **0.193** | **0.241** | 0.195 | 0.243 | 0.212 | 0.265 | 0.217 | 0.261 | 0.214 | 0.270 | 0.462 | 0.437 | 0.219 | 0.263 | 0.325 | 0.347 |
| 1st Count | | **34** | | 11 | | 0 | | 3 | | 0 | | 0 | | 0 | | 0 | |

Table 2: Univariate forecasting results for the M4 dataset. We report the weighted average forecasting error from all four subsets and full results are provided in Appendix D.1.

| | Models | **Koopa** | N-HiTS | N-BEATS | PatchTST | TimesNet | DLinear | MICN | KNF | FiLM | Autoformer |
|---|---|---|---|---|---|---|---|---|---|---|---|
| Weighted Average | sMAPE | **11.863** | 11.960 | 11.910 | 13.022 | 11.930 | 12.418 | 13.023 | 12.126 | 12.489 | 14.057 |
| | MASE | **1.595** | 1.606 | 1.613 | 1.814 | 1.597 | 1.656 | 1.836 | 1.641 | 1.690 | 1.954 |
| | OWA | **0.858** | 0.861 | 0.862 | 0.954 | 0.867 | 0.891 | 0.960 | 0.874 | 0.902 | 1.029 |

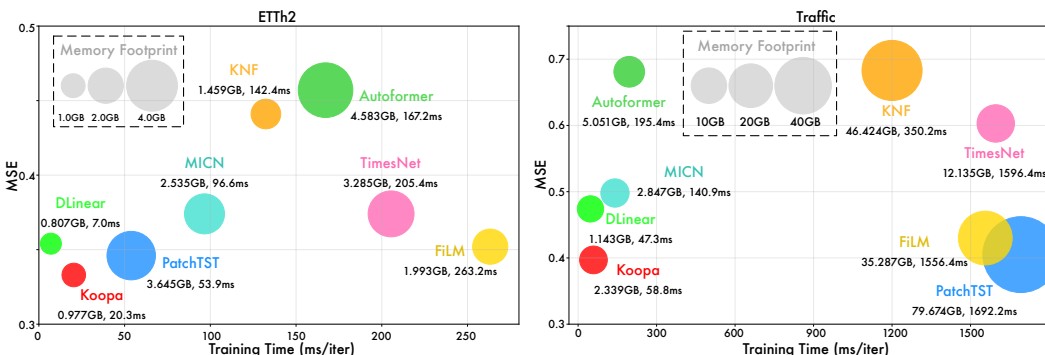

Figure 4: Model efficiency comparison. The performance comes from Table 1 with forecast window length $H = 144$. Training time and memory footprint are recorded with the same batch size and official code configuration. Full results of all six datasets are provided in Appendix D.3.

**Model efficiency**    We comprehensively evaluate the model efficiency from three aspects: forecasting performance, training speed, and memory footprint. In Figure 4, we compare the efficiency under two representative datasets with different variate numbers (7 in ETTh2 and 862 in Traffic).

Compared with the state-of-the art forecasting model PatchTST, Koopa saves **62.3%** and **96.5%** training time respectively in the ETTh2 and Traffic datasets with only **26.8%** and **2.9%** memory footprint. Concretely, the averaged training time and memory ratio of Koopa compared to PatchTST are **22.7%** and **24.0%** in all six datasets (see Appendix D.3 for the detail). Besides, as an efficient MLP-based forecaster, Koopa is also capable of learning nonlinear dynamics from time-variant and time-invariant components, and thus achieves a better performance.

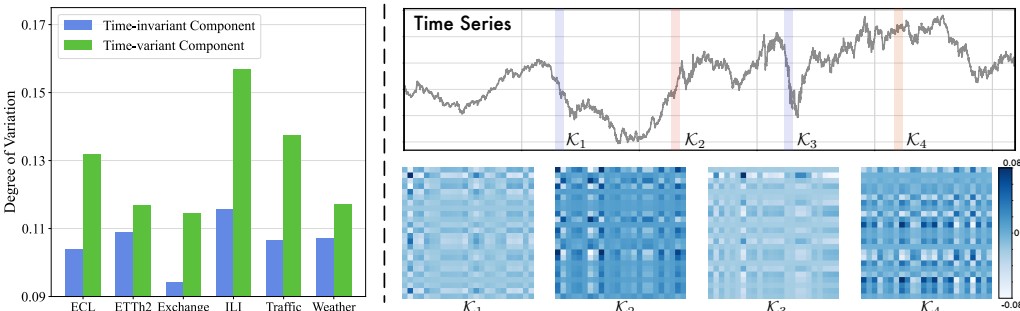

Figure 5: Left: Comparison of Degree of Variation (the standard deviation of linear weighting fitted on different periods), we plot respective values of disengaged components on all six datasets. Right: A case of localized Koopman operators calculated on the Exchange dataset at the interval of one year.

## 5.2    Model Analysis

**Dynamics disentanglement**    To validate the disentangling effect of our proposed Fourier Filter, we divide the whole time series into 20 subsets of different periods and conduct respective linear regression on the components disentangled by Fourier Filter. The standard deviation of the linear weighting reflects the variation of point-to-point temporal dependencies, which works as the manifestation of time-variant property. We plot the value as *Degree of Variation* (Figure 5 Left). It can be observed that larger deviations occur in the time-variant component, which indicates the proposed module successfully disentangles two types of dynamics from the perspective of frequency domain.

**Case study**    We present a case study on real-world time series (exchange rate) on the right of Figure 5. We sample the lookback window at the interval of one year and visualize the Koopman operators calculated in Time-variant KP. It can be clearly observed that localized operators can exhibit changing temporal patterns in different periods, indicating the necessity of utilizing varying operators to describe time-variant dynamics. And interpretable insights are also presented as series uptrends correspond to heatmaps with large value and downtrends are reflected with small value.

**Ablation study**  We conduct ablations on Koopa. As shown in Table 3, Time-variant and Time-invariant KPs perform as complementary modules to explore the dynamics underlying time series, and discarding any one of them will lead to the inferior performance. Besides, we evaluate alternative decomposition filters to disentangle time series dynamics. We find the proposed Fourier Filter conducts effective disentanglement, where the amplitude statistics of frequency spectrums from different periods are utilized to exhibit time-agnostic information. Therefore, Koopa tackling the right dynamics with complementary modules can achieves the best performance.

Table 3: Model ablation. *Only $K_{inv}$* uses one-block Time-invariant KP; *Only $K_{var}$* stacks Time-variant KPs only; *Truncated Filter* replaces Fourier Filter with High-Low Pass Filter; *Branch Switch* changes the order of KPs on disentangled components. The averaged results are listed here.

| Dataset | ECL | | ETTh2 | | Exchange | | ILI | | Traffic | | Weather | |
|---|---|---|---|---|---|---|---|---|---|---|---|---|
| Metric | MSE | MAE | MSE | MAE | MSE | MAE | MSE | MAE | MSE | MAE | MSE | MAE |
| Only $K_{inv}$ | 0.148 | 0.250 | 0.312 | 0.358 | 0.120 | 0.241 | 2.146 | 0.963 | 0.740 | 0.446 | 0.170 | 0.213 |
| Only $K_{var}$ | 1.547 | 0.782 | 0.371 | 0.405 | 0.205 | 0.316 | 2.370 | 1.006 | 0.947 | 0.544 | 0.180 | 0.232 |
| Truncated Filter | 0.155 | 0.255 | 0.311 | 0.362 | 0.129 | 0.246 | 1.988 | 0.907 | 0.536 | 0.334 | 0.172 | 0.220 |
| Branch Switch | 0.696 | 0.393 | 0.344 | 0.385 | 0.231 | 0.325 | 2.130 | 0.964 | 0.451 | 0.304 | 0.173 | 0.221 |
| **Koopa** | **0.146** | **0.246** | **0.303** | **0.356** | **0.111** | **0.230** | **1.734** | **0.862** | **0.419** | **0.293** | **0.162** | **0.211** |

**Avoiding rigorous reconstruction**  Unlike previous Koopman Autoencoders, the proposed Koopman Predictor does not reconstruct the whole dynamics at once, but aims to portray the partial dynamics evolution. Thus we remove the reconstruction branch, which is only utilized during training in previous KAEs. In our deep residual structure, the predictive objective function works as a good optimization indicator. We validate the design in Table 4, where the performance of sorely forecasting objective optimized model is better than with an additional reconstruction loss. Because the end-to-end forecasting objective helps to reduce the optimization gap between training and inference, making it a valuable contribution of applying Koopman operators on end-to-end time series forecasting.

Table 4: Performance comparison of the dynamics learning blocks implemented by our proposed Koopman Predictor (*Koopa*) and the canonical Koopman Autoencoder [29](*KAE*).

| Dataset | ETTh2 | | Exchange | | ECL | | Traffic | | Weather | | ILI | |
|---|---|---|---|---|---|---|---|---|---|---|---|---|
| Model | MSE | MAE | MSE | MAE | MSE | MAE | MSE | MAE | MSE | MAE | MSE | MAE |
| **Koopa** | **0.303** | **0.356** | **0.110** | **0.230** | **0.143** | **0.243** | **0.404** | **0.277** | **0.161** | **0.210** | **1.734** | **0.862** |
| KAE | 0.312 | 0.361 | 0.129 | 0.248 | 0.169 | 0.269 | 0.463 | 0.329 | 0.170 | 0.217 | 2.189 | 0.974 |
| Promotion | 2.88% | | 14.73% | | 15.38% | | 12.74% | | 5.29% | | 20.79% | |

**Learning stable operators**  We turn to analyze our architectural design from the spectral perspective. The eigenvalues of the operator determine the amplitude of dynamics evolution. As most of non-stationary time series experience the distribution shift and can be regarded as an unstable evolution, the learned Koopman operator with the modulus far from the unit circle will cause non-divergent and even explosive trending in the long term, leading to training failures.

To tackle this problem generally faced by Koopman-based forecasters, we propose to utilize the disentanglement and deep residual structure. We measure the stability of the operator as the average distance of eigenvalues from the unit circle. As shown in Figure 6, the operator can become more stable by the above two techniques. The disentanglement helps to describe complex dynamics based on the decomposition and appropriate inductive bias can be applied. The architecture where each block is employed to fill the residual of the previously fitted dynamics reduces the difficulty of directly reconstructing complicated dynamics. Each block portrays the basic process driven by a stable operator within its power, which can be aggregated for a complicated non-stationary process.

## 5.3 Scaling Up Forecast Horizon

Most deep forecasting models work as a settled function once trained (e.g. input-$T$-output-$H$). For scenarios where the prediction horizon is mismatched or long-term, it poses two challenges for the trained model: (1) reuse parameters learned from observed series; (2) utilize incoming ground truth for model adaptation. The practical scenarios, which we name as *scaling up forecast horizon*, may

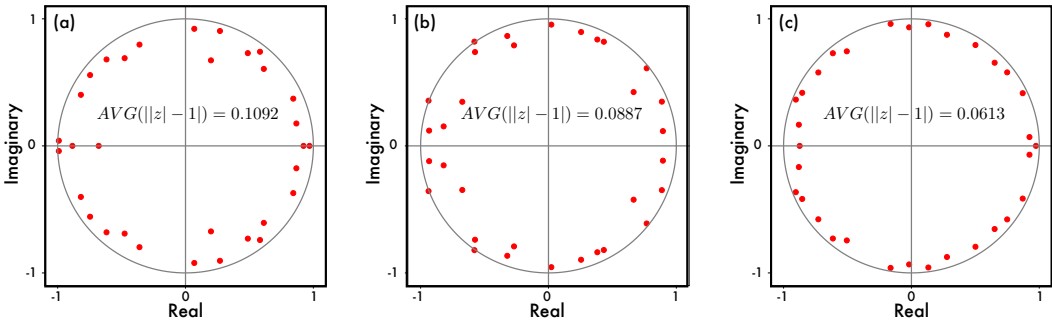

Figure 6: Visualization of the operator stability on the highly non-stationary Exchange dataset. We plot the first block time-invariant operator eigenvalues of the following design: (a) Single-block model with only time-invariant operator. (b) Single-block model with time-invariant and time-variant operators. (c) Two-block model with time-invariant and time-variant operators.

lead to failure on most deep models but can be naturally tackled by Koopa. In detail, we first train Koopa with forecast length $H_{\text{tr}}$ and attempt to apply it on a larger forecast length $H_{\text{te}}$.

**Method**   Koopa scales up forecast horizon as follows: Since Time-invariant KP has learned the globally shared dynamics and Time-variant KP can calculate localized operator $K_{\text{var}}$ within the lookback window, we freeze the parameters of trained Koopa but only use the incoming ground truth to adapt $K_{\text{var}}$. The naïve implementation uses incremental Koopman embedding with dimension $D$ and conducts Equation 10 to obtain an updated operator, which has a complexity of $\mathcal{O}(H_{\text{te}}D^3)$. We further propose an iterative algorithm with improved $\mathcal{O}((H_{\text{te}} + D)D^2)$ complexity. The detailed method implementations and complexity analysis can be found in Appendix A.

**Results**   As shown in Table 5, the proposed operator adaption mechanism further boosts the performance on the scaling up scenario, which can be attributed to more accurately fitted time-variant dynamics with incoming ground truth snapshots. Besides, the promotion becomes more significant when applied to non-stationary datasets (manifested as large ADF Test Statistic [10]).

Table 5: Scaling up forecast horizon: $(H_{\text{tr}}, H_{\text{te}}) = (24, 48)$ for ILI and $(H_{\text{tr}}, H_{\text{te}}) = (48, 144)$ for others. *Koopa* conducts vanilla rolling forecast and *Koopa OA* further introduces operator adaptation.

| Dataset | Exchange | | ETTh2 | | ILI | | ECL | | Traffic | | Weather | |
|---|---|---|---|---|---|---|---|---|---|---|---|---|
| ADF Test Statistic | (-1.889) | | (-4.135) | | (-5.406) | | (-8.483) | | (-15.046) | | (-26.661) | |
| Metric | MSE | MAE | MSE | MAE | MSE | MAE | MSE | MAE | MSE | MAE | MSE | MAE |
| Koopa | 0.214 | 0.348 | 0.437 | 0.429 | 2.836 | 1.065 | 0.199 | 0.298 | 0.709 | 0.437 | 0.237 | 0.276 |
| **Koopa OA** | **0.172** | **0.319** | **0.372** | **0.404** | **2.427** | **0.907** | **0.182** | **0.271** | **0.699** | **0.426** | **0.225** | **0.264** |
| Promotion (MSE) | 19.6% | | 14.9% | | 14.1% | | 8.5% | | 1.4% | | 5.1% | |

## 6   Conclusion

This paper tackles time series as dynamical systems. With disentangled time-variant and time-invariant components from non-stationary series, the Koopa model reveals the complicated dynamics hierarchically and leverages MLP modules to learn Koopman embedding and operator. Experimentally, our model shows competitive performance with remarkable efficiency and the potential to scale up the forecast length by operator adaptation. In the future, we will explore Koopa with the dynamic modes underlying non-stationary data using the toolbox of Koopman spectral analysis.

## Acknowledgments

This work was supported by the National Key Research and Development Plan (2021YFC3000905), National Natural Science Foundation of China (62022050 and 62021002), Beijing Nova Program (Z201100006820041), and BNRist Innovation Fund (BNR2021RC01002).

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

## A    Scaling Up Forecast Horizon

In this section, we introduce the capability of Koopa to scale up forecast horizon. In detail, we train a Koopa model with forecast length $H_{\text{tr}}$ and attempt to apply it on a larger length $H_{\text{te}}$. The basic approach conducts rolling forecast by taking the model prediction as the input of the next iteration until the desired forecast horizon is all filled. Instead, we further assume that after the model gives a prediction, the model can utilize the incoming ground truth for *model adaptation* and continue rolling forecast for the next iteration. It is notable that we do not retrain parameters during model adaptation, since it will lead to overfitting on the incoming ground truth and Catastrophic Forgetting [11, 18, 35].

Koopa can naturally cope with the scenario by learning Koopman embedding and operator $K_{\text{inv}}$ in Time-invariant KPs while calculating localized operator $K_{\text{var}}$ to describe the dynamics in the temporal neighborhood. Therefore, we freeze the parameters of Koopa but only use the incoming ground truth for operator adaptation of $K_{\text{var}}$ in Time-variant KPs.

### A.1    Implementation of Operator Adaptation

At the beginning of rolling forecast, the Encoder in Time-variant KP outputs $D$-dimensional Koopman embedding for each observed series segment as $[z_1, z_2, \ldots, z_F]$, where $F = \frac{H_{\text{tr}}}{S}$ is the segment number with $S$ as the segment length. The operator $K_{\text{var}}$ in Time-variant KP is calculated as follows:

$$Z_{\text{back}} = [z_1, z_2, \ldots, z_{F-1}], \; Z_{\text{fore}} = [z_2, z_3, \ldots, z_F], \; K_{\text{var}} = Z_{\text{fore}}Z_{\text{back}}^{\dagger}, \tag{15}$$

where $Z_{\text{back}}, Z_{\text{fore}} \in \mathbb{R}^{D \times (F-1)}, K_{\text{var}} \in \mathbb{R}^{D \times D}$. With the calculated operator, we obtain the next predicted Koopman embedding by one-step forwarding:

$$\hat{z}_{F+1} = K_{\text{var}}z_F. \tag{16}$$

After decoding the embedding $\hat{z}_{F+1}$ to the series prediction, we can utilize the true value of incoming Koopman embedding $z_{F+1}$ obtained by Koopa with frozen parameters. Instead of using $K_{\text{var}}$ to obtain the next embedding

$\hat{z}_{F+2}$, we use incremental embedding collections $Z_{\text{back}+}, Z_{\text{fore}+} \in \mathbb{R}^{D \times F}$ to obtain a more accurate operator $K_{\text{var}+} \in \mathbb{R}^{D \times D}$ to describe the local dynamics:

$$Z_{\text{back}+} = [Z_{\text{back}}, z_F], \ Z_{\text{fore}+} = [Z_{\text{back}}, z_{F+1}], \ K_{\text{var}+} = Z_{\text{fore}+} Z_{\text{back}+}^{\dagger}. \tag{17}$$

The procedure repeats for $L$ times ($L \propto H_{\text{te}}$) until the forecast horizon is all filled, we formulate it as Algorithm 1. And experimental results (Koopa OA) in Section 5.3 have demonstrated the promotion of forecasting performance due to more precisely fitted dynamics.

## A.2 Computational Acceleration

The naïve implementation shown in Algorithm 1 repeatedly conducts Equation 17 on the incremental embedding collection to obtain new operators, which has a complexity of $\mathcal{O}(LD^3)$. We propose an equivalent algorithm with improved complexity of $\mathcal{O}((L+D)D^2)$ as shown in Algorithm 2.

**Theorem.** *Algorithm 2 gives the same $K_{var}$ as Algorithm 1 in each iteration with $\mathcal{O}(D^2)$ complexity.*

**Proof.** We start with the first iteration analysis. By the definition of Moore–Penrose inverse, we have $Z_{\text{back}}^{\dagger} Z_{\text{back}} = I_{F-1}$, where $I_{F-1}$ is an identity matrix with the dimension of $F-1$. When the model receives the incoming embedding $z_{F+1}$, incremental embedding $m = z_F, n = z_{F+1}$ will be appended to $Z_{\text{back}}$ and $Z_{\text{fore}}$ respectively. Instead of calculating new $K_{\text{var}+}$ from incremental collections, we utilize calculated $K_{\text{var}}$ to find the iteration rule on $K_{\text{var}+}$. Concretely, we suppose

$$Z_{\text{back}+}^{\dagger} = \begin{bmatrix} Z_{\text{back}}^{\dagger} - \Delta \\ b^{\top} \end{bmatrix} \in \mathbb{R}^{F \times D}, \tag{18}$$

where $\Delta \in \mathbb{R}^{(F-1) \times D}, b \in \mathbb{R}^D$ are variables to be identified. By the definition of Moore–Penrose inverse, we have $Z_{\text{back}+}^{\dagger} Z_{\text{back}+} = I_F$. By unfolding it, we have the following equations:

$$\Delta Z_{\text{back}} = \mathbf{0}, \ b^{\top} Z_{\text{back}} = \vec{0}, \ b^{\top} m = 1, \ Z_{\text{back}}^{\dagger} m - \Delta m = \vec{0}. \tag{19}$$

We suppose $\Delta = \delta b^{\top}$, where $\delta \in \mathbb{R}^{F-1}$, such that when $b^{\top} Z_{\text{back}} = \vec{0}$, then $\Delta Z_{\text{back}} = \mathbf{0}$. Then we have $Z_{\text{back}}^{\dagger} m - \delta b^{\top} m = Z_{\text{back}}^{\dagger} m - \delta = \vec{0}$, thus $\Delta = Z_{\text{back}}^{\dagger} m b^{\top}$. Given equations that $b^{\top} Z_{\text{back}} = \vec{0}$ and $b^{\top} m = 1$, we have the analytical solution of $b$:

$$b = r/||r||^2, \text{ where } r = m - Z_{\text{back}} Z_{\text{back}}^{\dagger} m. \tag{20}$$

Therefore, we find the equation between the incremental version $K_{\text{var}+}$ and calculated $K_{\text{var}}$:

$$Z_{\text{back}+}^{\dagger} = \begin{bmatrix} Z_{\text{back}}^{\dagger}(I_D - m b^{\top}) \\ b^{\top} \end{bmatrix}, \ K_{\text{var}+} = Z_{\text{fore}+} Z_{\text{back}+}^{\dagger} = K_{\text{var}} + (n - K_{\text{var}} m) b^{\top}, \tag{21}$$

where $m, n$ are the incremental embedding of $Z_{\text{back}}, Z_{\text{fore}}$ and $b$ can be calculated by Equation 20. We also derive the iteration rule on $X = Z_{\text{back}} Z_{\text{back}}^{\dagger}$ to obtain $b$, which is formulated as follows:

$$X_+ = Z_{\text{back}+} Z_{\text{back}+}^{\dagger} = X + (m - Xm) b^{\top} = X + r b^{\top}. \tag{22}$$

By adopting Equation 21–22 and permuting the matrix multiplication order, we reduce the complexity of each iteration to $\mathcal{O}(D^2)$. Therefore, Algorithm 2 has a overall complexity of $\mathcal{O}((L+D)D^2)$. Since $L \propto H_{\text{te}}$, Algorithm 1–2 have $\mathcal{O}(H_{\text{te}} D^3)$ and $\mathcal{O}((H_{\text{te}} + D)D^2)$ complexity respectively.

# B   Implementation Details

Koopa is trained with L2 loss and optimized by ADAM [17] with an initial learning rate of 0.001 and batch size set to 32. The training process is early stopped within 10 epochs. We repeat each experiment three times with different random seeds to obtain average test MSE/MAE and detailed results with standard deviations are listed in Table 6. Experiments are implemented in PyTorch [34] and conducted on NVIDIA TITAN RTX 24GB GPUs.

All the baselines that we reproduced are implemented based on the benchmark of TimesNet [47] Repository, which is fairly built on the configurations provided by each model's original paper or official code. Since several baselines adopt Series Stationarization from Non-stationary Transformers [28] while others do not, we equip all models with the method for a fair comparison.

# C   Hyperparameter Sensitivity

Considering the efficiency of hyperparameters search, we fix the segment length $S = T/2$ and the number of Koopa blocks $B = 3$ in all our experiments. We verify the robustness of Koopa of other hyperparameters as

---

**Algorithm 1** Koopa Operator Adaptation.

---

**Require:** Observed embedding $Z = [z_1, \ldots, z_F]$ and successively incoming ground truth embedding $[z_{F+1}, \ldots, z_{F+L}]$ with each embedding $z_i \in \mathbb{R}^D$.

1: $Z_{\text{back}} = [z_1, \ldots, z_{F-1}], Z_{\text{fore}} = [z_2, \ldots, z_F]$        $\triangleright Z_{\text{back}}, Z_{\text{fore}} \in \mathbb{R}^{D \times (F-1)}$

2: $K_{\text{var}} = Z_{\text{fore}} Z_{\text{back}}^{\dagger}$        $\triangleright K_{\text{var}} \in \mathbb{R}^{D \times D}$

3: $\hat{z}_{F+1} = K_{\text{var}} n$        $\triangleright \hat{z}_{F+1} \in \mathbb{R}^D$

4: **for** $l$ **in** $\{1, \ldots, L\}$**:**        $\triangleright z_{F+l}$ comes successively

5:      $m = z_{F+l-1}, n = z_{F+l}$        $\triangleright m, n \in \mathbb{R}^D$

6:      $Z_{\text{back}} \leftarrow [Z_{\text{back}}, m], Z_{\text{fore}} \leftarrow [Z_{\text{fore}}, n]$        $\triangleright Z_{\text{back}}, Z_{\text{fore}} \in \mathbb{R}^{D \times (F+l-1)}$

7:      $K_{\text{var}} = Z_{\text{fore}} Z_{\text{back}}^{\dagger}$        $\triangleright K_{\text{var}} \in \mathbb{R}^{D \times D}$

8:      $\hat{z}_{F+l+1} = K_{\text{var}} n$        $\triangleright \hat{z}_{F+l+1} \in \mathbb{R}^D$

9: **End for**

10: **Return** $[\hat{z}_{F+1}, \ldots, \hat{z}_{F+L+1}]$        $\triangleright$ Return predicted embedding

---

**Algorithm 2** Accelerated Koopa Operator Adaptation.

---

**Require:** Observed embedding $Z = [z_1, \ldots, z_F]$ and successively incoming ground truth embedding $[z_{F+1}, \ldots, z_{F+L}]$ with each embedding $z_i \in \mathbb{R}^D$.

1: $Z_{\text{back}} = [z_1, \ldots, z_{F-1}], Z_{\text{fore}} = [z_2, \ldots, z_F]$        $\triangleright Z_{\text{back}}, Z_{\text{fore}} \in \mathbb{R}^{D \times (F-1)}$

2: $K_{\text{var}} = Z_{\text{fore}} Z_{\text{back}}^{\dagger}, X = Z_{\text{back}} Z_{\text{back}}^{\dagger}$        $\triangleright K_{\text{var}}, X \in \mathbb{R}^{D \times D}$

3: $\hat{z}_{F+1} = K_{\text{var}} n$        $\triangleright \hat{z}_{F+1} \in \mathbb{R}^D$

4: **for** $l$ **in** $\{1, \ldots, L\}$**:**        $\triangleright z_{F+l}$ comes successively

5:      $m = z_{F+l-1}, n = z_{F+l}$        $\triangleright m, n \in \mathbb{R}^D$

6:      $r = m - Xm$        $\triangleright r \in \mathbb{R}^D$

7:      $b = r/||r||^2$        $\triangleright b \in \mathbb{R}^D$

8:      $K_{\text{var}} \leftarrow K_{\text{var}} + (n - K_{\text{var}} m) b^{\top}$        $\triangleright K_{\text{var}} \in \mathbb{R}^{D \times D}$

9:      $X \leftarrow X + r b^{\top}$        $\triangleright X \in \mathbb{R}^{D \times D}$

10:      $\hat{z}_{F+l+1} = K_{\text{var}} n$        $\triangleright \hat{z}_{F+l+1} \in \mathbb{R}^D$

11: **End for**

12: **Return** $[\hat{z}_{F+1}, \ldots, \hat{z}_{F+L+1}]^{\top}$        $\triangleright$ Return predicted embedding

---

Table 6: Detailed performance of Koopa. We report the MSE/MAE and standard deviation of different forecast horizons $\{H_1, H_2, H_3, H_4\} = \{24, 36, 48, 60\}$ for ILI and $\{48, 96, 144, 192\}$ for others.

| Dataset | ECL | | ETTh2 | | Exchange | |
|---|---|---|---|---|---|---|
| Horizon | MSE | MAE | MSE | MAE | MSE | MAE |
| $H_1$ | 0.130±0.003 | 0.234±0.003 | 0.226±0.003 | 0.300±0.003 | 0.042±0.002 | 0.143±0.003 |
| $H_2$ | 0.136±0.004 | 0.236±0.005 | 0.297±0.004 | 0.349±0.004 | 0.083±0.004 | 0.207±0.004 |
| $H_3$ | 0.149±0.003 | 0.247±0.003 | 0.333±0.004 | 0.381±0.003 | 0.130±0.005 | 0.261±0.003 |
| $H_4$ | 0.156±0.004 | 0.254±0.003 | 0.356±0.005 | 0.393±0.004 | 0.184±0.009 | 0.309±0.005 |

| Dataset | ILI | | Traffic | | Weather | |
|---|---|---|---|---|---|---|
| Horizon | MSE | MAE | MSE | MAE | MSE | MAE |
| $H_1$ | 1.621±0.008 | 0.800±0.006 | 0.415±0.003 | 0.274±0.005 | 0.126±0.005 | 0.168±0.004 |
| $H_2$ | 1.803±0.040 | 0.855±0.020 | 0.401±0.005 | 0.275±0.004 | 0.154±0.006 | 0.205±0.003 |
| $H_3$ | 1.768±0.015 | 0.903±0.008 | 0.397±0.004 | 0.276±0.003 | 0.172±0.005 | 0.225±0.005 |
| $H_4$ | 1.743±0.040 | 0.891±0.009 | 0.403±0.007 | 0.284±0.009 | 0.193±0.003 | 0.241±0.004 |

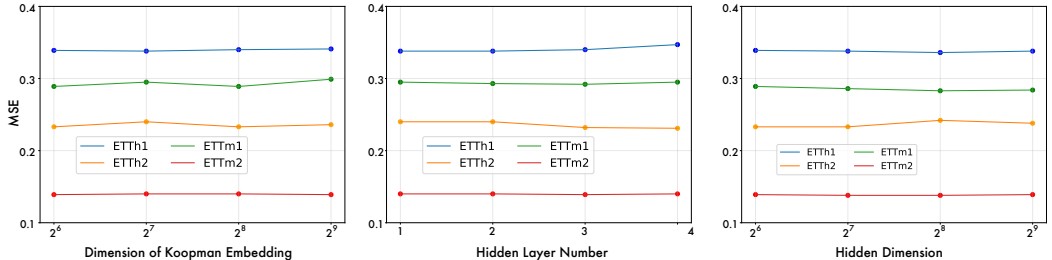

Figure 7: Left: Hyperparameter sensitivity with respect to the dimension of Koopman embedding, hidden layer number, and hidden dimension of Encoder and Decoder in Koopa.

follows: the dimension of Koopman embedding $D$, the hidden layer number $l$ and the hidden dimension $d$ used in Encoder and Decoder. As is shown in Figure 7, we find the proposed model is insensitive to the choices of above hyperparameters, which can be beneficial for practitioners to reduce tuning burden.

Intuitively, a larger dimension of Koopman embedding $D$ can bring about a lower approximation error. We further dive into it and find that stacking blocks can enhance the model capacity, and thus the performance is insensitive to $D$ when the model is deep enough. To address the concern, we further check the sensitivity of $D$ under varying block number $B$ in Figure 8. It can be seen that a larger $B$ generally leads to lower error even if $D$ is small. And the performance can be sensitive to $D$ when the model is not deep enough ($B = 1, 2$).

Besides, we find the proposed model can be insensitive to $S$ on several datasets while sensitive on ECL and Traffic datasets (The difference is about 10%). There are many variables in these two datasets, but our current design shares the $S$ for all variables. Since different variables with distinct evolution periods implicitly require different optimal $S$, the performance of the dataset with more variables is more likely to be influenced by $S$. Therefore, we set $S = T/2$ with relatively small performance fluctuation to deal with most situations.

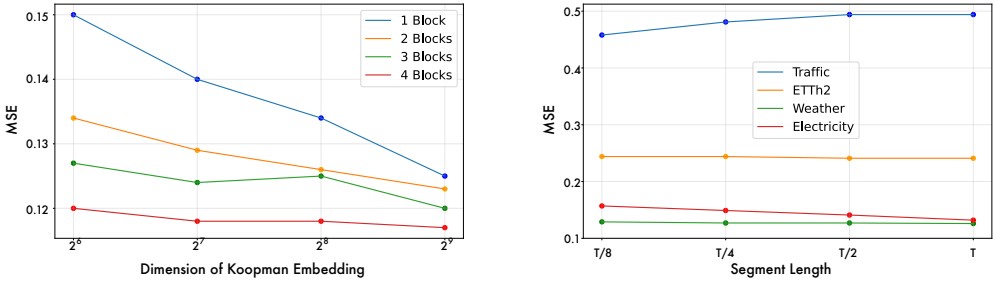

Figure 8: Left: Hyperparameter sensitivity of the dimension of Koopman embedding under different settings of the block number. Right: Hyperparameter sensitivity of the segment length.

## D    Supplementary Experimental Results

### D.1    Full Forecasting Results

Due to the limited pages, we list additional multivariate benchmarks on ETT datasets [53] in Table 7, which includes the hourly recorded ETTh2 and 15-minutely recorded ETTm1/ETTm2, and the full univariate results of M4 [39] in Table 8, which contains the yearly, quarterly and monthly collected univariate marketing data. Notably, Koopa still achieves competitive performance compared with state-of-the-art deep forecasting models and specialized univariate models.

### D.2    Full Ablation Results

We elaborately conduct model ablations to verify the necessity of each proposed module: Time-invariant KP, Time-variant KP, Fourier Filter and evaluate alternative choices to disentangle dynamics. As shown in Table 9, Koopa conducts effective disentanglement and tackles the right dynamics with complementary KPs, and thus achieves the best forecasting performance.

Table 7: Forecasting results with different forecast window lengths $H \in \{48, 96, 144, 192\}$ on ETT dataset. We set the lookback window length $T = 2H$.

| Models | | **KooPA** | | PatchTST [31] | | TimesNet [47] | | DLinear [51] | | MICN [43] | | KNF [44] | | FiLM [54] | | Autoformer [48] | |
|---|---|---|---|---|---|---|---|---|---|---|---|---|---|---|---|---|---|
| Metric | | MSE | MAE | MSE | MAE | MSE | MAE | MSE | MAE | MSE | MAE | MSE | MAE | MSE | MAE | MSE | MAE |
| ETTm1 | 48 | **0.283** | **0.333** | 0.286 | 0.336 | 0.308 | 0.354 | 0.322 | 0.355 | 0.294 | 0.353 | 1.026 | 0.792 | 0.324 | 0.353 | 0.592 | 0.419 |
| | 96 | **0.294** | **0.345** | 0.299 | 0.346 | 0.329 | 0.370 | 0.309 | 0.346 | 0.306 | 0.364 | 0.957 | 0.782 | 0.311 | 0.346 | 0.493 | 0.469 |
| | 144 | **0.322** | 0.366 | 0.325 | 0.363 | 0.358 | 0.387 | 0.327 | **0.359** | 0.342 | 0.390 | 0.921 | 0.760 | 0.328 | 0.358 | 0.735 | 0.569 |
| | 192 | **0.337** | 0.378 | 0.343 | 0.375 | 0.462 | 0.441 | **0.337** | **0.365** | 0.386 | 0.415 | 0.896 | 0.731 | 0.339 | 0.366 | 0.592 | 0.506 |
| ETTm2 | 48 | 0.134 | **0.226** | 0.135 | 0.231 | 0.142 | 0.234 | 0.144 | 0.240 | **0.131** | 0.238 | 0.621 | 0.623 | 0.146 | 0.243 | 0.191 | 0.280 |
| | 96 | **0.171** | **0.254** | 0.171 | 0.255 | 0.187 | 0.269 | 0.172 | 0.256 | 0.197 | 0.295 | 1.535 | 1.012 | 0.174 | 0.257 | 0.241 | 0.311 |
| | 144 | 0.206 | 0.280 | 0.205 | 0.282 | 0.216 | 0.291 | **0.200** | **0.276** | 0.210 | 0.297 | 1.337 | 0.876 | 0.204 | 0.279 | 0.300 | 0.352 |
| | 192 | 0.226 | 0.298 | 0.221 | 0.294 | 0.243 | 0.313 | **0.219** | **0.290** | 0.248 | 0.328 | 1.355 | 0.908 | 0.224 | 0.293 | 0.324 | 0.370 |
| ETTh1 | 48 | **0.336** | 0.377 | 0.337 | 0.375 | 0.365 | 0.399 | 0.343 | **0.371** | 0.375 | 0.406 | 0.876 | 0.709 | 0.407 | 0.427 | 0.442 | 0.438 |
| | 96 | **0.371** | 0.405 | 0.372 | **0.393** | 0.411 | 0.430 | 0.379 | **0.393** | 0.406 | 0.429 | 0.975 | 0.744 | 0.429 | 0.431 | 0.634 | 0.523 |
| | 144 | 0.405 | 0.418 | 0.394 | 0.412 | 0.442 | 0.447 | **0.393** | **0.403** | 0.437 | 0.448 | 0.801 | 0.662 | 0.451 | 0.448 | 0.522 | 0.491 |
| | 192 | 0.416 | 0.429 | 0.416 | 0.439 | 0.469 | 0.470 | **0.407** | **0.416** | 0.518 | 0.496 | 0.941 | 0.744 | 0.460 | 0.459 | 0.525 | 0.501 |

Table 8: Full univariate forecasting results for M4 dataset. We follow the same data processing and forecasting length settings used in TimesNet [47] benchmark. Additional forecasting models N-HiTS [7] and N-BEATS [32] are also included.

| Models | | **KooPA** | N-HiTS | N-BEATS | PatchTST | TimesNet | DLinear | MICN | KNF | FiLM | Autoformer |
|---|---|---|---|---|---|---|---|---|---|---|---|
| Year | sMAPE | **13.352** | 13.371 | 13.466 | 13.517 | 13.394 | 13.866 | 14.532 | 13.986 | 14.012 | 14.786 |
| | MASE | **2.997** | 3.025 | 3.059 | 3.031 | 3.004 | 3.006 | 3.359 | 3.029 | 3.071 | 3.349 |
| | OWA | **0.786** | 0.790 | 0.797 | 0.795 | 0.787 | 0.802 | 0.867 | 0.804 | 0.815 | 0.874 |
| Quarter | sMAPE | 10.159 | 10.454 | **10.074** | 10.847 | 10.101 | 10.689 | 11.395 | 10.343 | 10.758 | 12.125 |
| | MASE | 1.189 | 1.219 | **1.163** | 1.315 | 1.183 | 1.294 | 1.379 | 1.202 | 1.306 | 1.483 |
| | OWA | 0.895 | 0.919 | **0.881** | 0.972 | 0.890 | 0.957 | 1.020 | 0.965 | 0.905 | 1.091 |
| Month | sMAPE | **12.730** | 12.794 | 12.801 | 14.584 | 12.866 | 13.372 | 13.829 | 12.894 | 13.377 | 15.530 |
| | MASE | **0.953** | 0.960 | 0.955 | 1.169 | 0.964 | 1.014 | 1.082 | 1.023 | 1.021 | 1.277 |
| | OWA | 0.901 | 0.895 | **0.893** | 1.055 | 0.894 | 0.940 | 0.988 | 0.985 | 0.944 | 1.139 |
| Others | sMAPE | 4.861 | **4.696** | 5.008 | 6.184 | 4.982 | 4.894 | 6.151 | 4.753 | 5.259 | 5.841 |
| | MASE | **3.124** | 3.130 | 3.443 | 4.818 | 3.323 | 3.358 | 4.263 | 3.138 | 3.608 | 4.308 |
| | OWA | 1.004 | **0.988** | 1.070 | 1.140 | 1.048 | 1.044 | 1.319 | 1.019 | 1.122 | 1.294 |
| Weighted Average | sMAPE | **11.863** | 11.960 | 11.910 | 13.022 | 11.930 | 12.418 | 13.023 | 12.126 | 12.489 | 14.057 |
| | MASE | **1.595** | 1.606 | 1.613 | 1.814 | 1.597 | 1.656 | 1.836 | 1.641 | 1.690 | 1.954 |
| | OWA | **0.858** | 0.861 | 0.862 | 0.954 | 0.867 | 0.891 | 0.960 | 0.874 | 0.902 | 1.029 |

## D.3 Model Efficiency

We comprehensively compare the forecasting performance, training speed, and memory footprint of our model with well-acknowledged deep forecasting models. The results are recorded with the official model configuration and the same batch size. We visualize the model efficiency under all six multivariate datasets in Figure 9– 11. In detail, compared with the previous state-of-the-art model PatchTST [8], Koopa consumes only 15.2% training time and 3.6% memory footprint respectively in ECL, 37.8% training time and 26.8% memory in ETTh2, 23.5% training time and 37.3% memory in Exchange, 50.9% training time and 47.8% memory in ILI, 3.5% training time and 2.9% memory in Traffic, and 5.4% training time and 25.4% memory in Weather, leading to the averaged **77.3%** and **76.0%** saving of training time and memory footprint in all six datasets. The remarkable efficiency can be attributed to Koopa with MLPs as the building blocks, and we find the budget saving becoming more significant on datasets with more series variables (ECL, Traffic).

Besides, as an efficient linear model, the performance of Koopa still surpasses other MLP-based models. Especially, Compared with DLinear [51], our model reduces 38.0% MSE ($2.852 \rightarrow 1.768$) in ILI and 13.6% MSE ($0.452 \rightarrow 0.397$) in Weather. And the average MSE reduction of Koopa compared with the previous state-of-the-art MLP-based model reaches **12.2%**. Therefore, our proposed Koopa is efficiently built with MLP networks and shows great model capacity to exploit nonlinear dynamics and complicated temporal dependencies in real-world time series.

Table 9: Model ablation with detailed forecasting performance. We report forecasting results with different prediction lengths $\{24, 36, 48, 60\}$ for ILI and $H \in \{48, 96, 144, 192\}$ for others. For columns: Only $K_{inv}$ uses one-block Time-invariant KP; Only $K_{var}$ stacks Time-variant KPs only; *Truncated Filter* replaces Fourier Filter with High-Low Frequency Pass Filter; *Branch Switch* changes the order of KPs to deal with disentangled components.

| Models | | **KooPA** | | Only $K_{\text{inv}}$ | | Only $K_{\text{var}}$ | | Truncated Filter | | Branch Switch | |
|---|---|---|---|---|---|---|---|---|---|---|---|
| Metric | | MSE | MAE | MSE | MAE | MSE | MAE | MSE | MAE | MSE | MAE |
| ECL | 48 | **0.130** | **0.234** | 0.150 | 0.243 | 1.041 | 0.777 | 0.149 | 0.245 | 0.137 | 0.234 |
| | 96 | **0.136** | **0.236** | 0.137 | 0.242 | 4.643 | 1.669 | 0.172 | 0.280 | 2.240 | 0.724 |
| | 144 | **0.149** | 0.247 | 0.150 | 0.252 | 0.238 | 0.327 | **0.149** | **0.246** | 0.226 | 0.331 |
| | 192 | 0.156 | 0.254 | 0.158 | 0.260 | 0.267 | 0.355 | **0.152** | **0.248** | 0.181 | 0.284 |
| ETTh2 | 48 | **0.226** | **0.300** | 0.235 | 0.304 | 0.271 | 0.334 | 0.340 | 0.310 | 0.245 | 0.317 |
| | 96 | **0.297** | **0.349** | 0.311 | 0.353 | 0.382 | 0.405 | 0.301 | 0.352 | 0.343 | 0.384 |
| | 144 | **0.333** | 0.381 | 0.337 | **0.379** | 0.427 | 0.444 | 0.338 | 0.386 | 0.403 | 0.418 |
| | 192 | **0.356** | **0.393** | 0.363 | 0.397 | 0.402 | 0.437 | 0.363 | 0.400 | 0.384 | 0.420 |
| Exchange | 48 | **0.042** | **0.143** | 0.046 | 0.150 | 0.065 | 0.184 | 0.048 | 0.150 | 0.055 | 0.165 |
| | 96 | **0.083** | **0.207** | **0.083** | 0.210 | 0.147 | 0.274 | 0.087 | 0.210 | 0.151 | 0.277 |
| | 144 | **0.130** | **0.261** | 0.149 | 0.281 | 0.222 | 0.351 | 0.150 | 0.278 | 0.254 | 0.369 |
| | 192 | **0.184** | **0.309** | 0.200 | 0.322 | 0.385 | 0.456 | 0.229 | 0.345 | 0.463 | 0.490 |
| ILI | 24 | **1.621** | **0.800** | 2.165 | 0.882 | 1.972 | 0.919 | 2.140 | 0.874 | 2.092 | 0.894 |
| | 36 | **1.803** | 0.855 | 1.815 | 0.882 | 2.675 | 1.091 | 1.692 | **0.844** | 2.116 | 0.950 |
| | 48 | **1.768** | 0.903 | 2.107 | 0.981 | 2.446 | 1.045 | 1.762 | **0.895** | 2.394 | 1.084 |
| | 60 | **1.743** | **0.891** | 2.496 | 1.108 | 2.387 | 0.970 | 2.357 | 1.018 | 1.917 | 0.926 |
| Traffic | 48 | **0.415** | **0.274** | 0.445 | 0.295 | 0.915 | 0.536 | 0.668 | 0.363 | 0.468 | 0.300 |
| | 96 | **0.401** | **0.275** | 0.403 | 0.277 | 0.833 | 0.465 | 0.441 | 0.323 | 0.429 | 0.298 |
| | 144 | **0.397** | **0.276** | 0.400 | 0.278 | 0.816 | 0.452 | 0.436 | 0.321 | 0.438 | 0.307 |
| | 192 | **0.403** | **0.284** | 1.371 | 0.788 | 1.224 | 0.723 | 0.597 | 0.331 | 0.469 | 0.312 |
| Weather | 48 | 0.126 | 0.168 | 0.142 | 0.181 | 0.140 | 0.190 | **0.125** | **0.166** | 0.130 | 0.173 |
| | 96 | 0.154 | 0.205 | 0.164 | 0.209 | 0.169 | 0.224 | **0.154** | **0.202** | 0.163 | 0.210 |
| | 144 | **0.172** | **0.225** | 0.178 | 0.226 | 0.194 | 0.247 | 0.176 | 0.226 | 0.187 | 0.238 |
| | 192 | **0.193** | **0.241** | 0.195 | 0.245 | 0.217 | 0.268 | 0.195 | 0.244 | 0.212 | 0.261 |

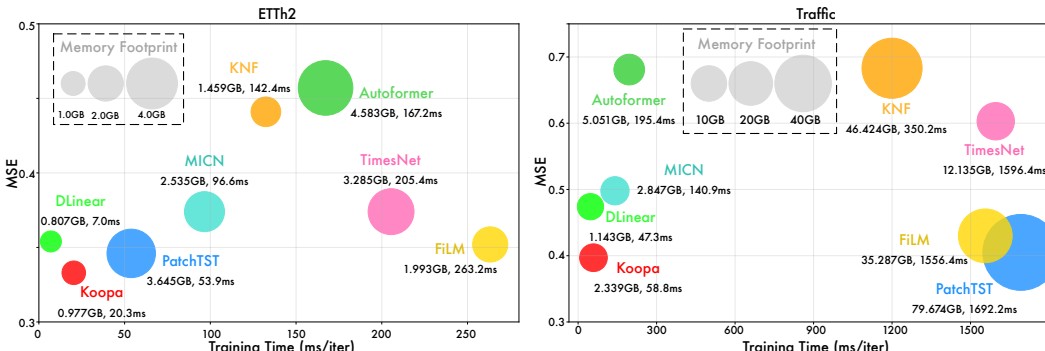

Figure 9: Model efficiency comparison with forecast length $H = 144$ for ETTh2 and Traffic.

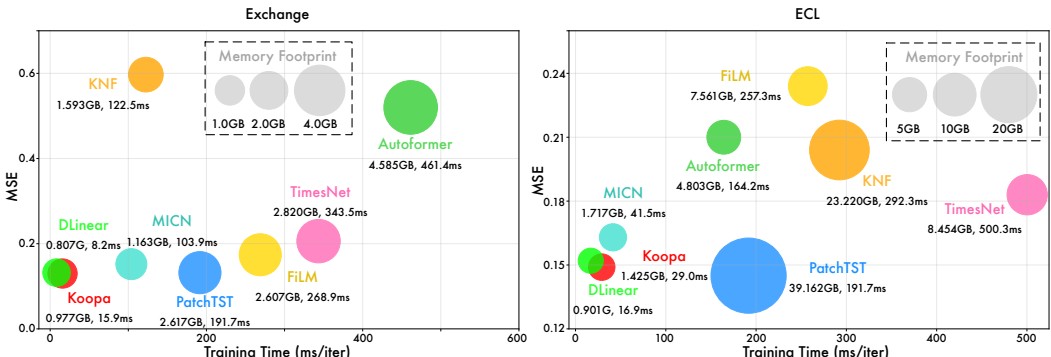

Figure 10: Model efficiency comparison with forecast length $H = 144$ for Exchange and ECL.

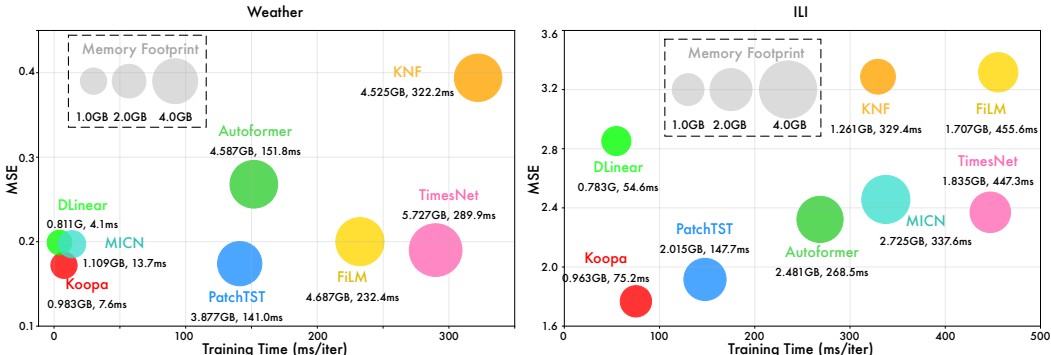

Figure 11: Model efficiency comparison with forecast length $H = 144$ for Weather and $48$ for ILI.

## D.4 Training Stability

Since the eigenvalues of the operator determine the amplitude of dynamics evolution, where modulus not close to one causes non-divergent and explosive evolution in the long-term, we highlight that a stable Koopman operator only describes weakly stationary series well. While most of the Koopman-based forecasters can suffer from the operator convergence problem induced by complicated non-stationary series variations, we employ several techniques to stabilize the training process.

**Operator initialization** We adopt the operator initialization strategy, where the operator starts from the multiplication of eigenfunctions with standard Gaussian distribution and all-one eigenvalues.

**Hierarchical disentanglement** In our model, each block learns weak stationary process hierarchically and feeds the residual of fitted dynamics for the next block to correct. Thus Koopman Predictor aims not to fully reconstruct the whole dynamics at once, but to partially describe dynamics, so rigorous reconstruction is not forced in each block, reducing the difficulty of portraying the non-stationary series as dynamics.

**Explosion checking** We introduce an explosion checking mechanism that replaces the operator encountering nan number with the identity matrix when the exponential multiplication of multiple time steps is detected.

Based on the proposed strategies, we provide the model training curves in Figure 12 to check the convergence of our proposed model and other forecasters. The training curves of the proposed model in blue demonstrate a consistent and smooth convergence, indicating its effectiveness in converging toward an optimal solution.

## E Broader Impact

### E.1 Impact on Real-world Applications

Our work copes with real-world time series forecasting, which is faced with intrinsic non-stationarity that poses fundamental challenges for deep forecasting models. Since previous studies hardly research the theoretical basis that can naturally address the time-variant property in non-stationary data, we propose a novel Koopman

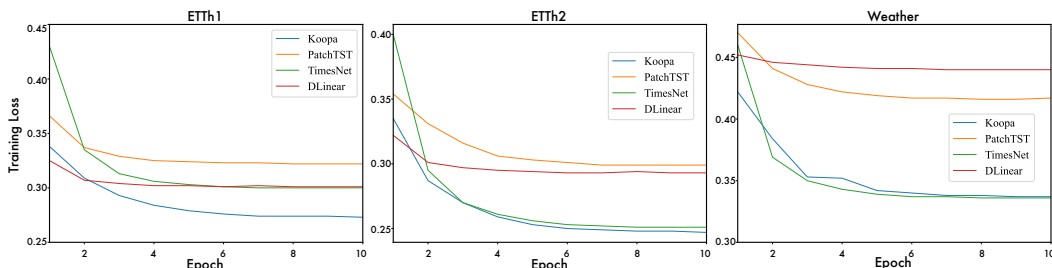

Figure 12: Training curves of different models on the ETT and Weather datasets.

forecaster that fundamentally considers the implicit time-variant and time-invariant dynamics based on Koopman theory. Our model achieves the state-of-the-art performance on six real-world forecasting tasks, covering energy, economics, disease, traffic, and weather, and demonstrates remarkable model efficiency in training time and memory footprint. Therefore, the proposed model makes it promising to tackle real-world forecasting applications, which helps our society prevent risks in advance and make better decisions with limited computational budgets. Our paper mainly focuses on scientific research and has no obvious negative social impact.

### E.2 Impact on Future Research

In this paper, we find modern Koopman theory natural to learn the dynamics underlying non-stationary time series. The proposed model explores complex non-stationary patterns with temporal localization inspired by Koopman approaches and implements respective deep network modules to disentangle and portray time-variant and time-invariant dynamics with the enlightenment of Wold's Theorem. The remarkable efficiency and insights from the theory can be instructive for future research.

## F Limitation

Our proposed model does not respectively considers dynamics in different variates, which leaves improvement for better multivariate forecasting with the consideration of various evolution patterns and series relationships. And Koopman spectral theory is still under leveraging in our work, which can discover Koopman modes to interpret the linear behavior underlying non-stationary data in a high-dimensional representation. Besides, Koopman theory for control considering factors outside the system can be promising for series forecasting with covariates, which leaves our future work.

