# OpenReview forum: "Koopa: Learning Non-stationary Time Series Dynamics with Koopman Predictors"
_NeurIPS.cc/2023/Conference — NeurIPS 2023 poster_

### Official Review · Reviewer_pXJ3 · 2023-06-14

**Soundness:** 2 fair
**Presentation:** 4 excellent
**Contribution:** 3 good
**Rating:** 6
**Confidence:** 5

**Summary:**

This paper tackles the task of non-stationary time series forecasting with Koopman operator. Specifically, the authors propose Koopa Blocks, which uses Fourier transform to disentangle the time series into time-variant and time-invariant components, and leverages Koopman
operator to model their dynamics respectively. The method is evaluated on six real-world benchmark datasets.

**Strengths:**

1. The paper is well-written and easy to follow.
2. This study represents a pioneering endeavor in integrating Fourier transform and Koopman theory, with the aim of disentangling time-variant and time-invariant components within the Fourier spectral domain. The idea is both innovative and thought-provoking, setting a precedent for future investigations into the research of deep Koopman methods.
3. The additional "Scaling Up Forecast Horizon" experiments demonstrate that Koopa is capable of accurately forecasting in situations involving length mismatches or long-term scenarios. In the realm of the Koopman method, these long-term prediction results are exceptional.
4. The proposed model is memory and time efficient compared to previous methods.

**Weaknesses:**

1. The performance improvement of univariate forecasting is incremental. Also, the improvement of multivariate forecasting is also not so significant under most cases.
2. The essence of the deep Koopman method lies in the spectrum of the Koopman operator, because the eigenvalues determine the model's behavior during long-term evolution. However, the author did not provide any analysis or visualized results regarding the eigenvalues. This lead to the following issues:
- Given the considerable forecasting length (up to 192 in this case), it is remarkable how the model manages to achieve accurate predictions without encountering issues such as explosion or decay during the forward pass. Is it true that all the eigenvalues of the system have a modulus of one, ensuring stable and non-divergent behavior throughout the prediction process?
- Intuitively, it would be expected that the time-variant Koopman operator possesses more eigenvalues with larger phase angles compared to the time-invariant operator. I wonder whether this is true in this case. What does the spectrum for these two operator looks like?



**Questions:**

1. The authors implement the DMD with torch.linalg.lstsq.  Were there any concerns regarding gradient explosion during the autograd process through this layer? Did the authors employ any techniques to stabilize the training process?

**Limitations:**

Yes.

---

> ### Author Rebuttal · Authors · 2023-08-09
>
>
> Many thanks to Reviewer pXJ3 for providing a detailed review and insightful questions.
>
> **Q1:** About the performance of the proposed method.
>
> To our best knowledge, the benchmark of time series forecasting has been excavated for a long time. And PatchTST as the SOTA forecasting model has surpassed concurrent work by a large margin. However, the Transformer-based model still suffers from the efficiency problem, which makes it less applicable for practitioners. Meanwhile, the recent revival of linear models presents a simple but effective approach, but the performance is still under promotion on non-stationary time series.
>
> Therefore, we delve for **applicable and efficient time series forecasting** based on Koopman theory and achieve **77.3%** training time and **76.0%** memory cut-off on average, while exceeding current SOTA in **34 out of 48** benchmarks. We hope it would help practitioners achieve performance on par with a heavily trained Transformer-based model while benefiting from the efficiency of linear models.
>
> **Q2**: Techniques for stabilizing the training process.
>
> As shown in our code implementation, we employ several techniques to stabilize the training process：
>
> - **Operator initialization:** We adopt the operator initialization as previous Koopman-based forecasters, where the eigenfunctions start from standard Gaussian distribution with all-one eigenvalues.
> - **Explosion checking**: When we find explosion occasionally happens in the proposed Koopman Predictor, we introduce an explosion checking mechanism that replacs the operator encountering nan with the identity matrix.
> - **Hierarchically disentanglement**: While reproducing the results of previous Koopman forecasters, we often find the training fails with explosive outputs. As the reviewer mentioned insightfully, we analyze the phenomenon and find the hierarchical disentanglement could alleviate the problem (Please refer to $\underline{\text{Q2 and Q4 of the reviewer M3Xz}}$ for the analysis).
>
> Besides, motivated by your suggestion of analyzing of the eigenvalues, we conduct spectral experiments to ablate the disentanglement and hierarchical forecasting mechanism. We use $average(||z|-1|)$ to measure the the stability of operator describing the pattern evolution. On Exchange dataset, we plot the eigenvalues $z$ and the measurement of the time-invariant operator in the first layer in the following cases:
>
> - a. Single-layer model with only time-invariant operator.
>
> - b. Single-layer model with time-invariant and time-variant operators.
>
> - c. Two-layer model with time-invariant and time-variant operators.
>
> The visualization results are shown in $\underline{\text{Figure 2 of the global response}}$, with the introduction of disentanglement and hierarchically stacking, we observed the eigenvalues become more close to the unit circle and present a more stable operator.
>
> **Q3**: Spectral analysis of time-invariant and time-variant operators.
>
> Thanks for this valuable suggestion. We visualize the eigenvalues of time-invariant and time-variant operators on Exchange dataset in $\underline{\text{Figure 4 of the global response}}$, And we have the following two findings:
>
> - We find it not obvious that more eigenvalues with larger phase angles exist in the time-variant Koopman operator than in the time-invariant operator. For example, in time series $y_t=\alpha sin(\omega_0 t)+\beta sin(\omega_t t) + \epsilon_t$, where $\alpha, \beta, \omega_0$ do not change with time, $\epsilon_t$ is white noise and $\omega_t$ varies window-wisely, the first term $\alpha sin(\omega_0 t)$ behaves time-invariantly because of stable periodicity, while $\beta sin(\omega_t t)$ with varying periodicity is close to our desired time-variant component. However, no partial order assumption is made between $\omega_0$ and $\omega_t$. Thus, the phase angles of eigenvalues may not have explicit relations.
> - Time-variant Koopman operators have many multiple eigenvalues around zero, which indicates simpler evolution patterns than Time-invariant operators. It's possible because Time-variant KP learns the dynamics within one lookback window, while the other learns the dynamics underlying the whole dataset.

---

> > ### Comment · Reviewer_pXJ3 · 2023-08-16
> >
> > Thanks for the response. While the concern regarding performance gain remains significant in the paper, I recognize that the proposed method presents an inspiring idea by combining Koopman theory and spectral methods. Moreover, its high interpretability adds value to the work, making it a valuable contribution to the Koopman learning community. I will raise my rating to weak accept.

---

> > > ### Author Response · Authors · 2023-08-17
> > > **Thanks for Your Response and Raising the Score**
> > >
> > > Many thanks to Reviewer pXJ3 for the insightful pre-rebuttal review and valuable feedback. Your detailed suggestions help us a lot in the rebuttal and paper revision!

---

### Official Review · Reviewer_EGYL · 2023-07-03

**Soundness:** 2 fair
**Presentation:** 3 good
**Contribution:** 3 good
**Rating:** 5
**Confidence:** 4

**Summary:**

This work propose a new architecture inspired by Koopman theory for time-series forecasting task. To address the non-stationarity problem, this work follow the idea of disentangle local and global time-series representation and forecasting model respectively. Particularly, this work propose to use Fourier filter on long sequence to extract more robust and stable patterns for global model and Koopa Block to hierarchically disentangle local and global representation in a residual structure. The proposed method achieves competitive performance when comparing with recent baselines.

**Strengths:**

1.  The structure is well organized. Figures 2 and 3 provide a clear visual representation of how the proposed method works.

2. The experiments are comprehensive. The proposed method can beat the recent method PatchTST on the commonly used long sequence forecasting benchmark.

**Weaknesses:**

1. Although the paper's title revolves around the Koopman theory, the proposed method is generic and not closely related to it.
From my understanding, most of technique contributions lie in how to extract local and global representation and design forecasting model for non-stationary time-series. The work solely follow the the conclusion of Koopman theory and model the latent dynamic with linear transition model.

2. Technique novelty is limited. The general idea of using local and global presentation/model to tackle non-stationary time-series tasks has been used. This work builds upon these foundations and introduces several modifications by integrating some other methods used in recent deep time series forecasting, like residual block, Fourier filter.

3.  Section 1 needs improvements. It is clear for me how Koopman operator address the non-linear dynamics, but section 1 does not clearly show how existing work of Koopman theory tackle non-stationary problems. Could the authors elaborate more on line 40-42? In line 52-54, could the authors explain the motivation of replacing reconstruction loss? In line 30-31, the authors introduce "Non-stationary  ...... in different periods". However, the proposed method do not show the capability to detect period boundary.

4. The experiments are comprehensive, but it would be better to design experiments revolve around the topic and the proposed method. Ablation studies can successfully validate the effectiveness of the design of local and global model, but the experiment associated time and memory cost analysis might lose focus. The authors can also explore the effect of number of block to verify the capability of hierarchical disentanglement.

5. It would be better to discuss the related work of non-stationary time series forecasting, like regime switching state-space models, time-varying state-space models, etc.

**Questions:**

how to set the hyperparameters  D?

**Limitations:**

N.A.

---

> ### Author Rebuttal · Authors · 2023-08-09
>
> Many thanks to Reviewer EGYL for the detailed and insightful review.
>
> **Q1:** Technique novelty of proposed method.
>
> As clarified in $\underline{\text{Q5 of the reviewer M3Xz}}$, we sum up "how existing work of Koopman theory tackles non-stationary problems" and highlight our difference with previous Koopman forecasters.
>
> And we'd like to clarify our work is not straightforward as "integrating some other methods used in recent deep time series forecasting, like residual block, Fourier Filter":
> - To our best knowledge, no previous Koopman forecasters consider stackable modular design and hierarchical forecast. To address the issue, we propose specialized Koopman Predictors to learn respective dynamics.
> - We make scalable incorporation with deep residual structure. It is totally different from residual connection proposed in ResNet. Instead, we feed **the residual of fitted input** of one layer to the next layer in the hope to learn hierarchical dynamics and **aggregate forecast** from all layers as the model output.
> - We propose a way to **extract the spectrum distribution of dataset**, which is based on FFT but is just the first step of our filter. Fourier Filter is not like canonical filters, which are implemented by truncating pre-defined frequency. Besides, the proposed filter is specially designed for Koopman forecasters, and the extraction of time-variant component is hardly explored in previous work.
>
> **Q2:** The motivation to stack Koopa Blocks and replace reconstruction.
>
> Thanks a lot for your valuable suggestion to verify the effect of block stacking, please refer to the results and our analysis in $\underline{\text{Q4 of the reviewer M3Xz}}$.
>
> Besides, with the introduction of hierarchical forecasting, the ground truth of reconstruction should be the residual of fitted dynamics, which is unavailable during training. Therefore, we adopt the deep residual structure, where solely forecasting objective works as a good optimization indicator, and we further validate it in $\underline{\text{Table 2 of the global response}}$.
>
> **Q3:** How does the proposed model detect the period boundary?
>
> Firstly, Fourier Filter helps the model detect window-shared periodicity. Based on the disentangled time-invariance component with almost shared periodicities, the global operator learns lookback-forecast **transition**, which relies less on the specific period boundary but more on the evolution pattern (such as how the phase changes and the decay rate of eigenfunctions).
>
> **Q4:** Reclarify the position and contribution of our work.
>
> The reviewer mentioned that our work "solely follows the conclusion of Koopman theory". We agree with this argument but also would like to clarify that the position of our work is to **propose an applicable architecture based on Koopman theory for real-world forecasting**. Concretely, our contributions mainly revolve around architecture and can be listed as follows:
> - Specialized designed disentanglement for Koopman forecasters.
> - Koopman Predictor with reconstruction loss raveled out and achieve end-to-end objective optimization.
> - Hierarchically disentangling and deep residual structure for model scale up.
> - Derive operator updating rule with reduced complexity based on the linear properties of Koopman operator.
>
> **Q5:** Discuss related works of non-stationary time series forecasting.
>
> Previous works adopt series stationarization to attenuate non-stationarity for better predictability, such as Adaptive Norm, DAIN, and ReVIN.
>
> Several recent works have also explored model architecture for non-stationary forecasting, especially with theory support that addresses time-variant properties. As the reviewer insightfully mentioned, the State Space Models (SSM) share lots of similarities with Koopman forecasters. Representatives such as Kalman Filters widely applied on control systems, and deep SSMs are good at modeling long sequences. Therefore, we'd like to discuss their similarity and differences as follows:
>
> (1) Similarities:
> - They portray the time series as the state transition of the system.
> - They are able to portray time-variant transitions.
> - They can introduce external driving factors to address non-stationarity caused by regime/concept drifting.
>
> (2) Differences:
> - Approach to describing time-variance: Regime Switching State Space Models assume pre-defined regime number and conditions to transit (Markov Process modeling). Time-variant SSM requires exquisitely designed transitions such as the HiPPO matrix. While our Koopman forecaster analyzes time-variant dynamics by disentanglement and leveraging DMD.
> - Model implementation: SSM is always implemented by RNN, with memory to model long-term dependencies. Canonical Koopman forecasters are always incorporated with AutoEncoder. Besides, the transition learner of Koopman forecasters (e.g. parameterized matrix) can be more simple and interpretable.
> - Objective optimization: SSM relies on Variational Bayesian Inference and always uses KL-divergency (CrossEntropy) for optimization. Koopman forecasters are traditionally optimized by reconstruction and dynamics advancing loss (MSE). And our model is solely optimized by end-to-end forecasting loss.
>
> To further address your concern, we also include the widely used SSM: LSSL and Regime Switching LSSL as additional baselines. Here are the results. Koopa still achieves the best performance in all benchmarks.
>
> |Average(MSE)|ECL|ETTh2|Exchange|ILI|Traffic|Weather|
> |-|-|-|-|-|-|-|
> |LSSL|0.284|1.426|1.036|4.580|0.699|0.216|
> |RegimeSwitching|0.164|0.361|0.176|2.841|0.495|0.189|
> |Koopa|**0.143**|**0.303**|**0.110**|**1.734**|**0.404**|**0.161**|
>
> **Q6:** Hyperparameter analysis of the dimension of Koopman embedding $D$.
>
> $D$ is selected from $\{64, 128, 256, 512\}$. We have also provided hyperparameter sensitivity analysis in $\underline{\text{Section 3 of supplementary materials}}$, where we find the performance insensitive to the choices of $D$.

---

> > ### Comment · Reviewer_EGYL · 2023-08-16
> >
> > - Could the authors clarify the connection between the proposed architecture design to the Koopman theory, how the proposed architecture can reduce approximation error of non-linear dynamic system? In theory, higher dimension dimension means lower approximation error w.r.t non-linear function. How to explain that the performance is insensitive to the choices of D?
> >
> > - Real-data is complicated, different datasets usually have different issues leading to performance bottlenecks. How to demonstrate that the performance improvement over the real-world datasets stems from Koopman theory via a better approximated non-linear function instead of deep architecture design (residual block, periodicity pattern extraction for this long-sequence forecasting benchmark, global and local disentanglement), whose effectiveness has been widely demonstrated in recent research work.
> >
> > - Can the authors elaborate more on my 3rd question in weakness?

---

> > > ### Author Response · Authors · 2023-08-17
> > > **Thanks for the Reviewer's Prompt Reply (Part 1)**
> > >
> > > Thanks a lot for your valuable prompt comments. We'd like to provide more elaborations on each point of your response.
> > >
> > > **1. How the proposed architecture reduces the approximation error of non-linear dynamics**
> > >
> > > Due to the complexity and non-divergent eigenvalues evolution of non-linear dynamics, it is always challenging to directly apply operator learning on non-stationary series as one dynamics, which motivates us to consider the following aspects of architecture.
> > >
> > > (a) Stackable blocks learning hierarchical dynamics
> > >
> > > To our best knowledge, we make Koopman-based forecasters "deep" for the first time. Each layer learns weak stationary process with well-natured operators for **stable training** (Please refer to $\underline{\text{Q2 of Reviewer pXJ3}}$ for our detailed analysis). Deeper layers learn the residual of previously fitted dynamics and aggregate layer-wise dynamics for the final forecast, which **enhances the model capacity** for learning complex non-linear dynamics. As shown in $\underline{\text{Table 1 of the global response}}$, the difficulty of dynamics approximation in one block can be larger than in multiple blocks.
> > >
> > > (b) Time-variant operators based on new disentanglement
> > >
> > > Inspired by Koopman theory portraying non-linear dynamics into sub-regions individually governed by local operators, we **associate dynamics sub-regions to varying windows** in time series and calculate time-variant operators by analytical method DMD. Though previous work also utilizes time-variant operators, they do not **explicitly design time-variant and invariant disentanglement** and **elaborate on respective dynamics modeling**. In our architectural ablation of $\underline{\text{Table 3 of the main text}}$, we validate the effectiveness of applying disentanglement, learning parameterized operator globally, and analytically calculating operators locally.
> > >
> > > **2. How to explain the performance is insensitive to the choices of D**
> > >
> > > Thanks a lot for your question with scientific rigor. As we clarified that stacking blocks could enhance model capacity, the sensitivity analysis of $D$ is conducted under a fixed number of Koopa blocks $B=3$, which can be large enough for dynamics modeling. To further address your concern, we further check the sensitivity of $D$ under varying block number $B$.
> > >
> > > | ETTh1 (MSE) | D=64  | D=128 | D=256 | D=512 |
> > > | ----------- | ----- | ----- | ----- | ----- |
> > > | B=1         | 0.400 | 0.396 | 0.393 | 0.390 |
> > > | B=2         | 0.393 | 0.392 | 0.389 | 0.388 |
> > > | B=3         | 0.388 | 0.387 | 0.386 | 0.386 |
> > > | B=4         | 0.385 | 0.385 | 0.385 | 0.384 |
> > >
> > > | Exchange (MSE) | D=64  | D=128 | D=256 | D=512 |
> > > | -------------- | ----- | ----- | ----- | ----- |
> > > | B=1            | 0.150 | 0.140 | 0.134 | 0.125 |
> > > | B=2            | 0.134 | 0.129 | 0.126 | 0.123 |
> > > | B=3            | 0.127 | 0.124 | 0.125 | 0.120 |
> > > | B=4            | 0.120 | 0.118 | 0.118 | 0.117 |
> > >
> > > It can be more obvious that a larger $D$ leads to lower error when $B$ is small. But stacking blocks as $B=3$ makes the performance insensitive to $D$. So we will include the findings and update hyperparameter sensitivity meticulously in the final version of our supplementary materials.

---

> > > ### Author Response · Authors · 2023-08-17
> > > **Thanks for the Reviewer's Prompt Reply (Part 2)**
> > >
> > > **3. How to demonstrate the performance improvement stems from Koopman theory via a better approximated non-linear function instead of deep architecture design.**
> > >
> > > To check how much the performance is improved by the Koopman theory-inspired building block. We compare Koopa with competitive baseline NBEATS[1] and its variants, which are all built on deep residual architecture with three available choices of the basic block.
> > >
> > > * Trend Block: MLP learning weighting on pre-defined polynomial basis $\{1, t, ..., t^p\}$.
> > > * Seasonal Block: MLP learning weighting on pre-defined Fourier series basis $\{cos(2\pi it), sin(2\pi it)\}$.
> > >
> > > * Generic Block: MLP learning point-wise weighting from lookback to forecast window.
> > >
> > > The basic blocks can be replaced by our Koopa Block, which is composed of Disentanglement (Fourier Filter) and Koopman Predictor (Enc+Dec+Operator). It is notable that NBEATS do not explicitly design disentanglement as well, so we leverage the well-acknowledged trend-seasonal disentanglement proposed by Autoformer[2]. We also trail on the periodicity selection of Fourier series in Seasonal Block based on the sample rate of datasets for a fair comparison.
> > >
> > > | Datasets (MSE)  | ECL       | ETTh2     | Exchange  | ILI       | Traffic   | Weather   |
> > > | - | - | - | - | - | - | - |
> > > | NBeats Generic Block     | 0.190     | 0.246     | 0.050     | 3.302     | 0.620     | 0.141     |
> > > | NBeats Trend Block   | 0.201     | 0.258     | 0.048     | 2.610     | 0.656     | 0.142     |
> > > | NBeats Seasonal Block   | 0.181     | 0.238     | 0.056     | 3.359     | 0.613     | 0.135     |
> > > | NBeats Decomp + Seasonal + Trend Block | 0.198     | **0.224** | 0.043     | 2.552     | 0.698     | 0.155     |
> > > | Koopa Block (Ours)   | **0.130** | 0.226     | **0.042** | **1.621** | **0.415** | **0.126** |
> > >
> > > The results demonstrate the significant performance gain brought by Koopman-based block (even with disentanglement) in ECL, ILI, and Traffic datasets. Notably, Koopman theory that employs varying localized operators is suitable to tackle distribution shift and the introduction of measurement function learning enhances the model capacity for non-linearity. It is also quite different from recent research works (global local decomp + respective modeling) that the localized part is still modeled by global-shared learnable parameters.
> > >
> > >
> > >
> > > **4. Elaborate more on 3rd question in weakness**
> > >
> > > (a) Existing works of Koopman theory tackling non-stationary problems.
> > >
> > > While Koopman theory has been widely incorporated with AutoEncoder for sequential modeling and forecasting [3, 4], **there are still few works that attend to the power of Koopman theory for non-stationary problems**. We sum up the pipeline as follows:
> > >
> > > - Introduce extra constraints to refine operator stability on non-stationary series, (e.g. PCL [5]). However, the operator can be intrinsically unstable learned from non-stationary data, since the temporal statistics and evolution regime can change greatly. So the predicted time series can be over-stationary.
> > > - Use local and global operators to deal with temporal distribution shifts. Since learning one operator for the whole dynamics is hard, KNF [6] as the most related work for non-stationary forecasting proposes to learn a unified global operator and utilize the self-attention map within the window as local operators. It directly **adds the global and local operator** as a time-variant transition, while we **add the components** given by respective predictors. It uses **classical AE structure** with reconstruction and forecasting loss and employs MLP for the coefficients on pre-defined measurement functions.
> > >
> > > (b) Elaborate more on lines 40-42.
> > >
> > > Take Duffing oscillator as shown on the left of $\underline{\text{Figure 1 of the main text}}$ for an example. It is hard to directly find one operator for portraying the dynamics but if we divide the dynamics into three localized sub-regions, we find several operators with simpler forms can be enough to describe. In that, for non-stationary time series with complicated time-variant $K_t$, **we discretize it into several window-wised operators** governing different periods (sub-regions).
> > >
> > > (c) Elaborate more on lines 52-54 and lines 30-31.
> > >
> > > As clarified in $\underline{\text{Q2 and Q3}}$, we hope our response can fulfilled the reviewer's expectations and would be very happy to answer any further detailed questions.
> > >
> > >
> > >
> > >
> > >
> > >
> > > [1] N-BEATS: Neural basis expansion analysis for interpretable time series forecasting.
> > >
> > > [2] Autoformer: Decomposition Transformers with Auto-Correlation for Long-Term Series Forecasting.
> > >
> > > [3] Deep learning for universal linear embeddings of nonlinear dynamics.
> > >
> > > [4] Learning deep neural network representations for koopman operators of nonlinear dynamical systems.
> > >
> > > [5] Forecasting sequential data using consistent koopman autoencoders.
> > >
> > > [6] Koopman neural forecaster for time series with temporal distribution shifts.

---

> > > ### Author Response · Authors · 2023-08-19
> > > **Looking forward to your reply.**
> > >
> > > We sincerely appreciate the time you dedicated to reviewing our paper. Given the limited timeframe for author-reviewer discussion, please kindly let us know if our response has addressed your concerns.
> > >
> > > Following your suggestions, we improve the paper in the following aspects:
> > >
> > > - We clarify how the proposed architecture reduces the approximation error of non-linear dynamics.
> > > - We further analyze the hyperparameter sensitivity of D.
> > > - We add comparisons with the deep architecture design and highlight the performance gain stemming from a better approximated non-linear dynamics with Koopman theory.
> > > - We elaborate on more on your 3rd question in weakness.
> > >
> > > Thanks again for your valuable review. Looking forward to your reply.

---

> > > ### Author Response · Authors · 2023-08-21
> > > **Request of Reviewer’s attention and feedback**
> > >
> > > Dear Reviewer EGYL,
> > >
> > > Thanks again for your valuable and constructive review, which has inspired us to improve our paper further substantially.
> > >
> > > Following your suggestions, we have clarified the connection between our architecture design to Koopman theory, further analyzed the hyperparameter sensitivity of $D$​, demonstrated the performance gain benefited from our Koopman-based blocks, and discussed all your mentioned weaknesses in every detail. We do our best to solve your concerns in the limited time and characters.
> > >
> > > We hope that this new version has addressed your concerns to your satisfaction. We eagerly await your reply and are happy to answer any further questions. We kindly remind you that the reviewer-author discussion phase will end by Aug 21st at 1 pm EDT, with just 4 hours left. After that, we may not have a chance to respond to your comments.
> > >
> > > Sincere thanks for your dedication!
> > >
> > > Authors

---

### Official Review · Reviewer_a2Yj · 2023-07-05

**Soundness:** 4 excellent
**Presentation:** 4 excellent
**Contribution:** 3 good
**Rating:** 8
**Confidence:** 2

**Summary:**

The authors address the issue that most real-world time-series are non-stationary. To tackle this problem they introduce Koopa, a globally learned (time-invariant) and localized (time-variant) linear Koopman operators, to exploit respective dynamics underlying different components. In practice, Koopa is composed of stackable Koopa Blocks, where each block learns operators hierarchically by taking the residual of previous block fitted dynamics as input. The authors evaluate the proposed model on variety of time-series data sets and compare their method with transformer and convolution based deep architectures. Koopa outperforms or performs on par with the existing methods, yet has a significantly lower computational costs. Moreover, the proposed method can be use in settings where the forecast horizon must be increased beyond the training set-up.

**Strengths:**

Originality:
The related work section in well written. It covers both relevant model types in the field, for (a) time series forecasting, TCN, RNN and transformers are mentioned, as well as the authors position the current work in its context; for (b) Koopman operators, the authors provide an overview of the related work and how their work differs.
The work proposes a novel method leveraging Fourier analysis to disentangle time-series components, eDMD for estimating time-variant Koopman Predictor, and time-invariant Koopman Prediction as a learnable parameter. It differentiates itself from KAEs due to its block like structure, where the residual of the previous block is fed as input into the next block.

Quality:
The submission is techinncally sound. The authors build upon Koopman operator theory and Wold's Theorem. In addition, experimentally they validate whether their proposed sub-components of Koopa, Fourier Filter, truly disentangles time-variant from time-invariant series. As well as provide an ablation on Koopa structure.

Clarity:
The work is well written providing sufficient relevant background knowledge on Koopman operator theory also for reader who is unfamiliar with this theory.

Significance:
The obtained results are significant. The proposed method outperforms or performs on par with existing time-series forecasting methods. More importantly, the method is very efficient with respect to training time and memory costs compared to transformer or convolution based methods. Lastly, the authors showcase that their method is applicable for scaling up forecast horizon, where standard deep learning architectures fail. In all, based on the submission the author's suggest a very promising modelling technique, however, I am not an expert of Koopman operator theory (nor the works in this sub-domain).

**Weaknesses:**

Clarity:
When first mentioned, I would recommend using the complete name of the model type rather than only the acronym, for example, temporal convolutional network rather than just TCN, it makes it easier for the reader to read.


**Questions:**

For time-variant KP how do you choose S, is the Time-variant KP sensitive to S?

**Limitations:**

The authors have not addressed limitation of their work. I would suggest the authors to include such section in the final version of their work.

---

> ### Author Rebuttal · Authors · 2023-08-09
>
> Many thanks to Reviewer a2Yj for providing thorough detailed comments.
>
> **Q1:** Hyperparameter sensitivity of the segment length $S$.
>
> As per your suggestion, we have checked the robustness of the proposed method with respect to the hyperparameter $S$, which varies in $\{H/8,H/4,H/2,H\}$, where the forecast length is $H$ and the lookback length $T=2H$.
>
> | segment length $S$ | Exchange  | Traffic   | ETTh2     | ILI       | Weather   | ECL       |
> | :----------------- | :-------- | :-------- | :-------- | --------- | --------- | --------- |
> | $H/8$              | 0.046     | **0.458** | 0.244     | **2.326** | 0.129     | 0.157     |
> | $H/4$              | **0.044** | 0.481     | 0.244     | 2.351     | 0.127     | 0.149     |
> | $H/2$              | 0.045     | 0.494     | **0.241** | 2.354     | 0.127     | 0.141     |
> | $H$                | 0.047     | 0.494     | **0.241** | 2.331     | **0.126** | **0.132** |
>
> We find the proposed model insensitive to $S$ on several datasets but sensitive on ECL and Traffic datasets. It may be because there are many variables in these two datasets, but we currently share $S$ on all variables. Since the prediction of some variables may be greatly affected by $S$, the forecasting performance of the dataset with more variables can be more sensitive to $S$.
>
>
>
> **Q2:** About the writing issues.
>
> Thanks for your valuable suggestions. We will replace the acronym of models with their complete names. All the changes will be included in the final version of our work.
>
> **Q3:** About the limitations of the proposed method.
>
> Thanks a lot for your concern, we evaluate the limitations of our proposed method in $\underline{\text{Section 6 of supplementary materials}}$, which can be summarized as follows:
>
> - The design space of the encoder and decoder. We will further consider the embedding of various evolution patterns for better multivariate forecasting and have deeper integration of Koopman measurement functions.
> - The model interpretability. We will dig into Koopman modes decomposition to reveal the linear behavior underlying non-stationary time series.
> - Incorporation with DMDc. We will consider factors outside observations to tackle time series forecasting with covariates.

---

> > ### Comment · Reviewer_a2Yj · 2023-08-12
> >
> > Thank you for the additional ablation experiment and for voicing the limitations of the presented work, I recommend the authors to include the limitations in the main paper rather than the supplementary material. I will keep my score as it is.

---

> > > ### Author Response · Authors · 2023-08-13
> > > **Thanks for your response**
> > >
> > > Thanks a lot again for your response and every effort on the review. We will include the limitations in the final version of the main paper.

---

### Official Review · Reviewer_M3Xz · 2023-07-06

**Soundness:** 2 fair
**Presentation:** 3 good
**Contribution:** 1 poor
**Rating:** 5
**Confidence:** 4

**Summary:**

This paper addresses non-stationary time series forecasting, in which case the temporal distribution of time series is changing over time. Koopman theory is introduced in this paper for its fundamental capacity on modeling dynamical systems and a Koopman forecaster called Koopa is proposed. Concretely, in this paper, time series is disentangled into time-variant and time-invariant components. For time-invariant component, Koopa introduces a learnable matrix as Koopman operator, and for time-variant component, Koopa leverages eDMD to find the best fitted matrix that advances forward the system. The proposed framework is technically sound but somewhat lacks novelty.

**Strengths:**

1. Non-stationary time series forecasting is challenging and it is reasonable to apply Koopman theory to tackle this task.
2. Disentangling time series into variant and invariant components is intuitive.
3. The methods of constructing Koopman operators for both time-variant and -invariant components are straightforward.

**Weaknesses:**

1. The proposed model lacks novelty. Both disentangling time series and applying Koopman theory are commonly used techniques in time series analysis.
2. The technical contribution of this paper is limited. The extraction of time variant and invariant components is too simple and lacks analysis. The application of Koopman theory is simply the construction of Koopman operators.
3. Hierarchically disentangling time series lacks intuition and motivation. What's the point of further disentangle variant component into invariant and variant components?
4. The experiment does not involve highly non-stationary datasets.
5. Koopman operators are highly correlated to measurement functions. In the branch of time variant component, although the model constructs diverse Koopman operators, the measurement, i.e., the encoder, is static. For more technical contributions, a deeper integration of Koopman's theory is needed.

**Questions:**

See weaknesses.

**Limitations:**

Not applicable.

---

> ### Author Rebuttal · Authors · 2023-08-09
>
> Many thanks to Reviewer M3Xz for providing a detailed review.
>
> **Q1:** Novelty of disentangling module.
>
> We'd like to emphasize our proposed method differs from previous works as follows:
> - It considers time-variant and invariant disentanglement, which is essential for Koopman forecasters, and few series decompositions specialize in it.
> - The module is not a traditional filter. It filters components by **spectrum distribution of the dataset**, instead of pre-defined frequency thresholds.
>
> Concretely, it collects statistics by FFT. The subset in the top percentage sorted by amplitude indicates **dominant time-invariant properties (e.g. periodicity) shared among the dataset**.
>
> We have conducted analysis in $\underline{\text{Figure 5 of the main text}}$, which presents the disentangling effects. And ablation studies in $\underline{\text{Table 3 of the main text}}$ check its necessity and compares the performance of other filter choices.
>
> **Q2:** Motivation of the disentanglement.
>
> Koopa is essentially based on disentanglement, which is **motivated by Wold's Theorem**. We sum up motivations as follows:
> - Wold's Theorem implies weakly stationary series can be decomposed into deterministic and stochastic parts. The former (e.g. sine wave) can be easily modeled by an operator, while the latter is the output of a linear filter with stationary processes as input.
> - We utilize data-dependant encoder to learn underlying stationary process and use time-variant operators to portray the varying behavior of the linear filter. The disentanglement is natural on series dynamics and has not been explored in previous works.
>
> As an intuitive example: $y_t=\alpha sin(\omega_0 t)+\beta sin(\omega_t t) + \epsilon_t$, where $\alpha, \beta, \omega_0$ do not change with time, $\epsilon_t$ is white noise and $\omega_t$ varies with window. $\alpha sin(\omega_0 t)$ behaves time-invariantly because of stable periodicity, while $\beta sin(\omega_t t)$ has varying periodicity. The components learned by Koopa are in $\underline{\text{Figure 3 of the global response}}$, we find that
> * Time-invariant KP predicts invariant components with stable period.
> * Time-variant KP can predict variant components with window-wise varying period.
>
> **Q3:** Experiments of highly non-stationary datasets.
>
> The reviewer mentioned it "does not involve highly non-stationary datasets." Based on ADF statistics, there is 99% confidence to accept that a unit root is presented, which indicated these datasets are highly non-stationary: Exchange, ETTh2, and ILI.
>
> We also include Cryptos evaluated in non-stationary forecaster KNF. Koopa still surpasses existing forecasting models.
>
> |MSE|Koopa|PatchTST| TimesNet| DLinear| KNF|
> |-| -| -|- | -| - |
> |Cryptos|**1.105**|1.111|1.110 |1.116|1.108|
>
> **Q4:** Motivation of the hierarchical disentanglement.
>
> The review questioned why "disentangle variant component into invariant and variant components". Instead, we **disentangle the residual of fitted time-variant component.**, which is motivated by the applicability of Koopman theory and scalability of deep model:
> - The operator eigenvalues determine long-term evolution, where modulus not close to one causes non-divergent evolution. We highlight **a stable Koopman operator only reconstruct well on weakly stationary series**. So we desire each layer to learn weak stationary process hierarchically and feed the residual of fitted dynamics for the next layer to correct.
> - Koopman Predictor aims not to fully reconstruct the whole dynamics at once, but to partially describe dynamics, so we do not force rigorous reconstruction in each layer.
> - Deep decomposition has been demonstrated as effective architecture (e.g.NBEATS, Autoformer) and makes the module scalable for complicated patterns.
>
> We show the effect of stacking in $\underline{\text{Table 1 of the global response}}$. Here are the observations:
> - Sinlge-layer model always leads to the worst forecasting, which is significant on non-stationary datasets (Exchange and ILI).
> - Stacking multi-layer Koopa blocks leads to increased performance on most datasets.
>
> **Q5:** Reclarify the contributions of applying Koopman theory on deep models.
>
> The reviewer stated "Koopman theory are commonly used techniques in time series analysis." We agree with the argument but want to highlight **incorporating theory into the architecture of deep models** is non-trival. We sum up previous works on Koopman forecasters as follows:
> - Utilize AutoEncoder to learn measurement functions.
> - Introduce extra constraints to refine operator stability on non-stationary series.
> - Use local and global operators to deal with temporal distribution shift.
>
> Instead, our work tackles the above as follows:
> - Design Koopman Predictor with reconstruction loss raveled out and incorporate it into deep residual structure.
> - Not employ any consistency constraint but exquisitely design hierarchal stackable blocks to refine stability.
> - Though previous work also utilizes respective operators, they do not explicitly design and apply time-variant and invariant disentanglement.
>
> Our work further copes with realistic problems that previous works have not explored:
> - End-to-end optimization: We observe in $\underline{\text{Table 2 of the global response}}$ that forecasting objective works as good indicator and end-to-end optimizing always achieves better forecasting.
> - Operator adaptation: We derive operator updating rule with reduced complexity for the first time and Koopa can utilize new snapshots to scale up forecast horizon.
>
> **Q6**: About the encoder design.
>
> Thanks for your suggestion for encoder design with deeper theory integration. We currently use MLP for the sake of efficiency. It still leaves an open problem and we have tried to make it not "static", such as learning the weighting on pre-defined measurements, but it does not achieve further performance improvement. We will appreciate it a lot if the reviewer could inspire us for other potential designs.

---

> > ### Comment · Reviewer_M3Xz · 2023-08-18
> >
> > Thanks for the efforts made on the response. The authors have clarified their motivations and emphasized the novelty. Additional experiment on Cryptos is included. Most of my concerns have been well addressed and I would like to raise my score to 5.

---

> > > ### Author Response · Authors · 2023-08-18
> > > **Thanks for Your Response and Raising the Score**
> > >
> > > We would like to thank Reviewer M3Xz for providing a detailed valuable pre-rebuttal review, which helps us a lot in the rebuttal and paper revision.
> > >
> > > Thanks again for your response and raising the score! In the fi nal version, we will elaborate more on clarifing motivations and novelty, and include the additional experiments to the paper.

---

### Official Review · Reviewer_36zV · 2023-07-06

**Soundness:** 3 good
**Presentation:** 3 good
**Contribution:** 3 good
**Rating:** 7
**Confidence:** 2

**Summary:**

The paper studied time series forecasting problem for weather forecasting, energy consumption, and financial assessment. To propose a model that generalizes on varying distribution, the authors proposed Koopa which is composed of modular Koopman predictors to describe and advance forward series dynamics. The motivation is Koopman-based methods are appropriate to learn non-stationary time series dynamics.
The implementation is straightforward in the supplement material. The experimental results show the effective performance of the proposed  methods compared with state-of-the-art methods.

**Strengths:**

The paper proposed a novel method to solve the time series forecasting problem. The introduction of Koopa Block is interesting. The experimental results show that the model outperforms existing methods with more efficient memory usage.

**Weaknesses:**

It is interesting to see more analysis about Koopa blocks. For example on Figure 5, authors showed the visualized blocks are different which corresponds to different curves. But it is clear that the curves of K1, K2, and K2 share the similarity that they goes down first, then goes up. It is interesting to see the correlation analysis if there is some.

The experiments show the proposed method achieved better performance with efficient training time and memory usage. I am not sue if there is convergence analysis and what are the parameters, such as the number of epochs for training, batch size.

**Questions:**

1. Is there any correlation between Koopa blocks if the curves share similarity as shown in Figure 5 ?
2. How is the convergence of the proposed method and compared methods?
3. What are the parameters, such as the number of epochs for training, batch size for experiments?

**Limitations:**

There is no theoretical analysis but the main contribution is to propose a novel architecture.

---

> ### Author Rebuttal · Authors · 2023-08-09
>
> Many thanks to Reviewer 36zV for providing thorough insightful comments.
>
> **Q1:** The convergence of the proposed method and compared methods.
>
> As shown in $\underline{\text{Figure 1 of the global response}}$, we provide the model training curves to check the convergence of our proposed model and other methods. The training curves of the proposed model in blue demonstrate a consistent and smooth convergence, indicating its effectiveness in converging towards an optimal solution.
>
> **Q2:** Analysis of learned operators in Koopa blocks.
>
> As shown in $\underline{\text{Figure 5 of the main text}}$, we have qualitatively revealed the correlation that non-contemporaneous subseries that differ in **global tendency** will be distinguished by the heatmap weight of operators. As the reviewer mentioned insightfully, they also share **similar series variations, but with varied durations**. We find it not easy to be directly visualized since it can be manifested by varying decay rates of respective evolution of eigenfunctions. It leaves instructive for us to improve model interpretability with further consideration of Koopman eigenfunctions analysis.
>
> Besides, for a quantitative correlation analysis, we sample subseries monthly from Exchange dataset and calculate the correlation between **global tendency** and **operators**:
>
> - We use linear regression to fit each subseries and use the slope as a manifestation of global tendency.
> - We consider several properties of learned operators: the sum of elements (Sum), the sum of eigenvalues (Trace), the maximum absolute value of eigenvalues (2-Norm), and the Frobenius norm (F-Norm).
>
> The Pearson correlation $\tau$ between slopes and operator properties is listed as follows:
>
> |        | Sum       | Trace | 2-Norm | F-Norm |
> | ------ | :---: | :---: | :---: | :---: |
> |   $\tau$   | **0.845** | 0.605 | 0.523  | 0.624  |
>
> We observe a strong correlation between the global tendency and the sum of elements. This suggests that there is indeed a correlation between Koopa blocks when their curves exhibit similarity.
>
> **Q3:** About the detailed training parameters.
>
> As per your request, we provide training parameter as follows, which is shared on all six benchmarks.
>
> | Batch Size | Training Epochs | Learning Rate | Early Stoping Patience | LR Decay Strategy                           |
> | :---: | :---:| :---:| :---: | :---: |
> | 32         | 10              | 0.001         | 3                      | ExponentialLR with decay rate: $\gamma=0.5$ |

---

> > ### Comment · Reviewer_36zV · 2023-08-21
> > **official comment from reviewer 36zV**
> >
> > Thanks for responses. The authors answer my questions. I will keep my score to support the paper.

---

### Author Rebuttal · Authors · 2023-08-09

##  Global Response to All Reviewers

We sincerely thank all the reviewers for their insightful reviews and valuable comments, which are instructive for us to improve our paper further.

To cope with non-stationary time series forecasting, this paper proposes Koopman-based forecasters (Koopa) to portray the underlying time-variant properties. Inspired by Koopman theory, we design renovated Koopman Predictors and disentanglement to hierarchically reveal complicated dynamics. We delve into architectural design for applicable forecasters on real-world non-stationary series, with end-to-end forecasting objective optimization, fairly increased model efficiency, and operator adaptation to scale up forecast horizon. **Comprehensive experiments and module analysis are included. Koopa achieves competitive performance while saving 77.3% training time and 76.0% memory on six real-world benchmarks.**

The reviewers generally held positive opinions of our paper, in that the proposed method is "**novel**", "**technically sound**", "**a pioneering endeavor**", and "**the idea is innovative and thought-provoking, setting a precedent for future investigations into the research of deep Koopman methods**", this paper is "**well-written**" and "**provides a clear visual representation** " and the experiments are "**comprehensive**", "**significant**".

The reviewers also raised insightful and constructive concerns. We made every effort to address all the concerns by providing sufficient evidence and requested results. Here is the summary of the major revisions:

- **Analysis of operators (Reviewer 36zV, pXJ3)**: We provide the correlation analysis between time series and learned operators. To address the issue of operator stability, we visualize the respective eigenvalues in Koopman Predictors. The results present the effectiveness of our architectural design for better convergence of training operators.
- **Motivation of proposed modules (Reviewer M3Xz)**: We illustrate our motivation in both theoretical and experimental aspects. We clarify disentanglement and block stacking based on Wold's Therom, Koopman theory and the scalability of deep model. By conducting ablation and evaluating the other possible designs, we verify that our model still achieves the best performance and good generality.
- **Technical novelty (Reviewer EGYL, M3Xz)**: We highlight our difference with previous works on decomposition and Koopman forecasting. We specifically design and apply the disentanglement for Koopman-based forecasters and renovate canonical KAEs with raveled-out reconstruction branch by incorporating it into deep residual structure.
- **Analysis of hyperparameters (Reviewer a2Yj, EGYLj)**: We newly add hyperparameters sensitivity analysis on the segment length and the dimension of Koopman embedding. And we also clarify the selection strategy of model hyperparameters.
- **Discussion of related works (Reviewer EGYL)**: We discuss the similarities and differences between our method and related works on non-stationary time series forecasting. We especially compare Koopa with State Spaces Models in the aspects of implementation, optimization, and the way to describe time-variance. We also conduct experiments on the full benchmarks to check their performance.

The valuable suggestions from reviewers are very helpful for us to revise the paper to a better shape. We'd be very happy to answer any further questions.

Besides, shared Tables of the global response are listed as follows, and Figures are provided in PDF.

> Table.1  Analysis of block number.
| Block Number (MSE) | ILI       | ETTm2     | Exchange  | Weather   | ETTh2     | ECL       |
| ------------------ | :-------- | :-------- | :-------- | :-------- | --------- | --------- |
| 1                  | 2.184     | 0.136     | 0.520     | 0.129     | 0.242     | 0.153     |
| 2                  | 1.980     | 0.133     | 0.506     | 0.126     | 0.241     | **0.141** |
| 3                  | 1.974     | 0.132     | 0.479     | 0.125     | **0.236** | 0.142     |
| 4                  | **1.850** | **0.131** | **0.473** | **0.124** | 0.239     | 0.158     |

> Table.2 Analysis of introducing reconstruction
| ETTh2 (MSE\|MAE)      | Predict 48             | Predict 96             | Predict 144            | Predict 192            |
| --------------------- | --------------- | --------------- | ---------------------- | ------------|
| Koopa                 | **0.226** \| **0.300** | **0.297** \| **0.349** | **0.333** \| **0.381** | 0.356 \| 0.393         |
| + Reconstruction | 0.244 \| 0.311         | 0.309 \| 0.356         | 0.339 \| 0.385         | **0.354** \| **0.392** |
||||||||
| **Exchange**   |             |             |             |            |
| Koopa                 | **0.042**\| **0.143** | **0.083** \| **0.207** | **0.130** \| **0.261** | **0.184** \| **0.309** |
| + Reconstruction | 0.047 \| 0.152        | 0.102 \| 0.225         | 0.131 \| 0.263         | 0.235 \| 0.352         |
||||||||
| **ECL**   |             |             |             |            |
| Koopa                 | **0.130** \| **0.234** | **0.136** \| **0.236** | **0.149** \| **0.247** | **0.156** \| **0.254** |
| + Reconstruction | 0.183 \| 0.282         | 0.159 \| 0.260         | 0.165 \| 0.265         | 0.167 \| 0.269         |
||||||||
| **Traffic**     |              |             |   |          |
| Koopa                 | **0.415** \| **0.274** | **0.401** \| **0.275** | **0.397** \| **0.276** | **0.403** \|**0.284** |
| + Reconstruction | 0.518 \| 0.353         | 0.446 \| 0.317         | 0.445 \| 0.325 | 0.444 \| 0.322        |
||||||||
| **Weather**     |              |             |           |           |
| Koopa                 | **0.126** \| **0.168** | **0.154** \| **0.205** | **0.172** \| **0.225** | **0.193** \| **0.241** |
| + Reconstruction | 0.143 \| 0.186         | 0.163 \| 0.208         | 0.178 \| 0.226         | 0.196 \| 0.246         |

---

### Decision · Program_Chairs · 2023-09-21

**Decision:**

Accept (poster)

**Comment:**

The authors propose a deep architecture for time series forecasting, inspired by the theory of Koopman operators. After a thorough discussion with many clarifications from the authors, the reviewers converged on a positive assessment of the work. While the connection to Koopman theory is somewhat tenuous, the architectural proposal is interesting and the results are promising.

AC note: In addition to addressing the reviewers' concerns and adding the promised improvements to the final manuscript, I encourage the authors to correct grammatical mistakes throughout the paper. For example, the first line of the abstract has a singular/plural error.